# First Order Constrained Optimization in Policy Space

**Yiming Zhang**
New York University
yiming.zhang@cs.nyu.edu

**Quan Vuong**
UC San Diego
qvuong@ucsd.edu

**Keith W. Ross**
New York University Shanghai
New York University
keithwross@nyu.edu

## Abstract

In reinforcement learning, an agent attempts to learn high-performing behaviors through interacting with the environment, such behaviors are often quantified in the form of a reward function. However some aspects of behavior—such as ones which are deemed unsafe and to be avoided—are best captured through constraints. We propose a novel approach called First Order Constrained Optimization in Policy Space (FOCOPS) which maximizes an agent's overall reward while ensuring the agent satisfies a set of cost constraints. Using data generated from the current policy, FOCOPS first finds the optimal update policy by solving a constrained optimization problem in the nonparameterized policy space. FOCOPS then projects the update policy back into the parametric policy space. Our approach has an approximate upper bound for worst-case constraint violation throughout training and is first-order in nature therefore simple to implement. We provide empirical evidence that our simple approach achieves better performance on a set of constrained robotics locomotive tasks.

## 1  Introduction

In recent years, Deep Reinforcement Learning (DRL) saw major breakthroughs in several challenging high-dimensional tasks such as Atari games (Mnih et al., 2013, 2016; Van Hasselt et al., 2016; Schaul et al., 2015; Wang et al., 2017), playing go (Silver et al., 2016, 2018), and robotics (Peters and Schaal, 2008; Schulman et al., 2015, 2017b; Wu et al., 2017; Haarnoja et al., 2018). However most modern DRL algorithms allow the agent to freely explore the environment to obtain desirable behavior, provided that it leads to performance improvement. No regard is given to whether the agent's behavior may lead to negative or harmful consequences. Consider for instance the task of controlling a robot, certain maneuvers may damage the robot, or worse harm people around it. RL safety (Amodei et al., 2016) is a pressing topic in modern reinforcement learning research and imperative to applying reinforcement learning to real-world settings.

Constrained Markov Decision Processes (CMDP) (Kallenberg, 1983; Ross, 1985; Beutler and Ross, 1985; Ross and Varadarajan, 1989; Altman, 1999) provide a principled mathematical framework for dealing with such problems as it allows us to naturally incorporate safety criteria in the form of constraints. In low-dimensional finite settings, an optimal policy for CMDPs with known dynamics can be found by linear programming (Kallenberg, 1983) or Lagrange relaxation (Ross, 1985; Beutler and Ross, 1985).

While we can solve problems with small state and action spaces via linear programming and value iteration, function approximation is required in order to generalize over large state spaces. Based on recent advances in local policy search methods (Kakade and Langford, 2002; Peters and Schaal, 2008; Schulman et al., 2015), Achiam et al. (2017) proposed the Constrained Policy Optimization (CPO) algorithm. However policy updates for the CPO algorithm involve solving an optimization problem through Taylor approximations and inverting a high-dimensional Fisher information matrix.

These approximations often result in infeasible updates which would require additional recovery steps, this could sometimes cause updates to be backtracked leading to a waste of samples.

In this paper, we propose the First Order Constrained Optimization in Policy Space (FOCOPS) algorithm. FOCOPS attempts to answer the following question: given some current policy, what is the best constraint-satisfying policy update? FOCOPS provides a solution to this question in the form of a two-step approach. First, we will show that the best policy update has a near-closed form solution when attempting to solve for the optimal policy in the nonparametric policy space rather than the parameter space. However in most cases, this optimal policy is impossible to evaluate. Hence we project this policy back into the parametric policy space. This can be achieved by drawing samples from the current policy and evaluating a loss function between the parameterized policy and the optimal policy we found in the nonparametric policy space. Theoretically, FOCOPS has an approximate upper bound for worst-case constraint violation throughout training. Practically, FOCOPS is extremely simple to implement since it only utilizes first order approximations. We further test our algorithm on a series of challenging high-dimensional continuous control tasks and found that FOCOPS achieves better performance while maintaining approximate constraint satisfaction compared to current state of the art approaches, in particular second-order approaches such as CPO.

## 2 Preliminaries

### 2.1 Constrained Markov Decision Process

Consider a Markov Decision Process (MDP) (Sutton and Barto, 2018) denoted by the tuple $(\mathcal{S}, \mathcal{A}, R, P, \mu)$ where $\mathcal{S}$ is the state space, $\mathcal{A}$ is the action space, $P : \mathcal{S} \times \mathcal{A} \times \mathcal{S} \to [0, 1]$ is the transition kernel, $R : \mathcal{S} \times \mathcal{A} \to \mathbb{R}$ is the reward function, $\mu : \mathcal{S} \to [0, 1]$ is the initial state distribution. Let $\pi = \{\pi(a|s) : s \in \mathcal{S}, a \in \mathcal{A}\}$ denote a policy, and $\Pi$ denote the set of all stationary policies. We aim to find a stationary policy that maximizes the expected discount return $J(\pi) := \mathbb{E}_{\tau \sim \pi} \left[ \sum_{t=0}^{\infty} \gamma^t R(s_t, a_t) \right]$. Here $\tau = (s_0, a_0, \ldots,)$ is a sample trajectory and $\gamma \in (0, 1)$ is the discount factor. We use $\tau \sim \pi$ to indicate that the trajectory distribution depends on $\pi$ where $s_0 \sim \mu$, $a_t \sim \pi(\cdot|s_t)$, and $s_{t+1} \sim P(\cdot|s_t, a_t)$. The value function is expressed as $V^\pi(s) := \mathbb{E}_{\tau \sim \pi} \left[ \sum_{t=0}^{\infty} \gamma^t R(s_t, a_t) \middle| s_0 = s \right]$ and action-value function as $Q^\pi(s, a) := \mathbb{E}_{\tau \sim \pi} \left[ \sum_{t=0}^{\infty} \gamma^t R(s_t, a_t) \middle| s_0 = s, a_0 = a \right]$. The advantage function is defined as $A^\pi(s, a) := Q^\pi(s, a) - V^\pi(s)$. Finally, we define the discounted future state visitation distribution as $d^\pi(s) := (1 - \gamma) \sum_{t=0}^{\infty} \gamma^t P(s_t = s|\pi)$.

A Constrained Markov Decision Process (CMDP) (Kallenberg, 1983; Ross, 1985; Altman, 1999) is an MDP with an additional set of constraints $\mathcal{C}$ which restricts the set of allowable policies. The set $\mathcal{C}$ consists of a set of cost functions $C_i : \mathcal{S} \times \mathcal{A} \to \mathbb{R}$, $i = 1, \ldots, m$. Define the $C_i$-*return* as $J_{C_i}(\pi) := \mathbb{E}_{\tau \sim \pi} [\sum_{t=0}^{\infty} \gamma^t C_i(s, a)]$. The set of feasible policies is then $\Pi_{\mathcal{C}} := \{\pi \in \Pi : J_{C_i}(\pi) \leq b_i, i = 1, \ldots, m\}$. The reinforcement learning problem w.r.t. a CMDP is to find a policy such that $\pi^* = \mathrm{argmax}_{\pi \in \Pi_{\mathcal{C}}} J(\pi)$.

Analogous to the standard $V^\pi$, $Q^\pi$, and $A^\pi$ for return, we define the cost value function, cost action-value function, and cost advantage function as $V_{C_i}^\pi$, $Q_{C_i}^\pi$, and $A_{C_i}^\pi$ where we replace the reward $R$ with $C_i$. Without loss of generality, we will restrict our discussion to the case of one constraint with a cost function $C$. However we will briefly discuss in later sections how our methodology can be naturally extended to the multiple constraint case.

### 2.2 Solving CMDPs via Local Policy Search

Typically, we update policies by drawing samples from the environment, hence we usually consider a set of parameterized policies (for example, neural networks with a fixed architecture) $\Pi_\theta = \{\pi_\theta : \theta \in \Theta\} \subset \Pi$ from which we can easily evaluate and sample from. Conversely throughout this paper, we will also refer to $\Pi$ as the *nonparameterized policy space*.

Suppose we have some policy update procedure and we wish to update the policy at the $k$th iteration $\pi_{\theta_k}$ to obtain $\pi_{\theta_{k+1}}$. Updating $\pi_{\theta_k}$ within some local region (i.e. $D(\pi_\theta, \pi_{\theta_k}) < \delta$ for some divergence

measure $D$) can often lead to more stable behavior and better sample efficiency (Peters and Schaal, 2008; Kakade and Langford, 2002; Pirotta et al., 2013). In particular, theoretical guarantees for policy improvement can be obtained when $D$ is chosen to be $D_{\mathrm{KL}}(\pi_\theta \| \pi_{\theta_k})$ (Schulman et al., 2015; Achiam et al., 2017).

However solving CMDPs directly within the context of local policy search can be challenging and sample inefficient since after each policy update, additional samples need to be collected from the new policy in order to evaluate whether the $C$ constraints are satisfied. Achiam et al. (2017) proposed replacing the cost constraint with a surrogate cost function which evaluates the constraint $J_C(\pi_\theta)$ using samples collected from the current policy $\pi_{\theta_k}$. This surrogate function is shown to be a good approximation to $J_C(\pi_\theta)$ when $\pi_\theta$ and $\pi_{\theta_k}$ are close w.r.t. the KL divergence. Based on this idea, the CPO algorithm (Achiam et al., 2017) performs policy updates as follows: given some policy $\pi_{\theta_k}$, the new policy $\pi_{\theta_{k+1}}$ is obtained by solving the optimization problem

$$\underset{\pi_\theta \in \Pi_\theta}{\text{maximize}} \quad \underset{\substack{s \sim d^{\pi_{\theta_k}} \\ a \sim \pi_\theta}}{\mathbb{E}} \left[ A^{\pi_{\theta_k}}(s,a) \right] \tag{1}$$

$$\text{subject to} \quad J_C(\pi_{\theta_k}) + \frac{1}{1-\gamma} \underset{\substack{s \sim d^{\pi_{\theta_k}} \\ a \sim \pi_\theta}}{\mathbb{E}} \left[ A_C^{\pi_{\theta_k}}(s,a) \right] \leq b \tag{2}$$

$$\bar{D}_{\mathrm{KL}}(\pi_\theta \| \pi_{\theta_k}) \leq \delta. \tag{3}$$

where $\bar{D}_{\mathrm{KL}}(\pi_\theta \| \pi_{\theta_k}) := \mathbb{E}_{s \sim d^{\pi_{\theta_k}}}[D_{\mathrm{KL}}(\pi_\theta \| \pi_{\theta_k})[s]]$. We will henceforth refer to constraint (2) as the *cost constraint* and (3) as the *trust region constraint*. For policy classes with a high-dimensional parameter space such as deep neural networks, it is often infeasible to solve Problem (1-3) directly in terms of $\theta$. Achiam et al. (2017) solves Problem (1-3) by first applying first and second order Taylor approximation to the objective and constraints, the resulting optimization problem is convex and can be solved using standard convex optimization techniques.

However such an approach introduces several sources of error, namely (i) Sampling error resulting from taking sample trajectories from the current policy (ii) Approximation errors resulting from Taylor approximations (iii) Solving the resulting optimization problem post-Taylor approximation involves taking the inverse of a Fisher information matrix, whose size is equal to the number of parameters in the policy network. Inverting such a large matrix is computationally expensive to attempt directly hence the CPO algorithm uses the conjugate gradient method (Strang, 2007) to indirectly calculate the inverse. This results in further approximation errors. In practice the presence of these errors require the CPO algorithm to take additional steps during each update in the training process in order to recover from constraint violations, this is often difficult to achieve and may not always work well in practice. We will show in the next several sections that our approach is able to eliminate the last two sources of error and outperform CPO using a simple first-order method.

## 2.3 Related Work

In the tabular case, CMDPs have been extensively studied for different constraint criteria (Kallenberg, 1983; Beutler and Ross, 1985, 1986; Ross, 1989; Ross and Varadarajan, 1989, 1991; Altman, 1999).

In high-dimensional settings, Chow et al. (2017) proposed a primal-dual method which is shown to converge to policies satisfying cost constraints. Tessler et al. (2019) introduced a penalized reward formulation and used a multi-timescaled approach for training an actor-critic style algorithm which guarantees convergence to a fixed point. However multi-timescaled approaches impose stringent requirements on the learning rates which can be difficult to tune in practice. We note that neither of these methods are able to guarantee cost constraint satisfaction during training.

Several recent work leveraged advances in control theory to solve the CMDP problem. Chow et al. (2018, 2019) presented a method for constructing Lyapunov function which guarantees constraint-satisfaction during training. Stooke et al. (2020) combined PID control with Lagrangian methods which dampens cost oscillations resulting in reduced constraint violations.

Recently Yang et al. (2020) independently proposed the Projection-Based Constrained Policy Optimization (PCPO) algorithm which utilized a different two-step approach. PCPO first finds the policy with the maximum return by doing one TRPO (Schulman et al., 2015) update. It then projects this policy back into the feasible cost constraint set in terms of the minimum KL divergence. While PCPO also satisfies theoretical guarantees for cost constraint satisfaction, it uses second-order approxima-

tions in both steps. FOCOPS is first-order which results in a much simpler algorithm in practice. Furthermore, empirical results from PCPO does not consistently outperform CPO.

The idea of first solving within the nonparametric space and then projecting back into the parameter space has a long history in machine learning and has recently been adopted by the RL community. Abdolmaleki et al. (2018) took the "inference view" of policy search and attempts to find the desired policy via the EM algorithm, whereas FOCOPS is motivated by the "optimization view" by directly solving the cost-constrained trust region problem using a primal-dual approach then projecting the solution back into the parametric policy space. Peters et al. (2010) and Montgomery and Levine (2016) similarly took an optimization view but are motivated by different optimization problems. Vuong et al. (2019) proposed a general framework exploring different trust-region constraints. However to the best of our knowledge, FOCOPS is the first algorithm to apply these ideas to cost-constrained RL.

## 3 Constrained Optimization in Policy Space

Instead of solving (1-3) directly, we use a two-step approach summarized below:

1. Given policy $\pi_{\theta_k}$, find an *optimal update policy* $\pi^*$ by solving the optimization problem from (1-3) in the nonparameterized policy space.
2. Project the policy found in the previous step back into the parameterized policy space $\Pi_\theta$ by solving for the closest policy $\pi_\theta \in \Pi_\theta$ to $\pi^*$ in order to obtain $\pi_{\theta_{k+1}}$.

### 3.1 Finding the Optimal Update Policy

In the first step, we consider the optimization problem

$$\underset{\pi \in \Pi}{\text{maximize}} \quad \underset{\substack{s \sim d^{\pi_{\theta_k}} \\ a \sim \pi}}{\mathbb{E}} \left[ A^{\pi_{\theta_k}}(s, a) \right] \tag{4}$$

$$\text{subject to} \quad J_C(\pi_{\theta_k}) + \frac{1}{1 - \gamma} \underset{\substack{s \sim d^{\pi_{\theta_k}} \\ a \sim \pi}}{\mathbb{E}} \left[ A_C^{\pi_{\theta_k}}(s, a) \right] \leq b \tag{5}$$

$$\bar{D}_{\text{KL}}(\pi \,\|\, \pi_{\theta_k}) \leq \delta \tag{6}$$

Note that this problem is almost identical to Problem (1-3) except the parameter of interest is now the nonparameterized policy $\pi$ and not the policy parameter $\theta$. We can show that Problem (4-6) admits the following solution (see Appendix A of the supplementary material for proof):

**Theorem 1.** *Let $\tilde{b} = (1 - \gamma)(b - \tilde{J}_C(\pi_{\theta_k}))$. If $\pi_{\theta_k}$ is a feasible solution, the optimal policy for (4-6) takes the form*

$$\pi^*(a|s) = \frac{\pi_{\theta_k}(a|s)}{Z_{\lambda,\nu}(s)} \exp\left( \frac{1}{\lambda} \left( A^{\pi_{\theta_k}}(s, a) - \nu A_C^{\pi_{\theta_k}}(s, a) \right) \right) \tag{7}$$

*where $Z_{\lambda,\nu}(s)$ is the partition function which ensures (7) is a valid probability distribution, $\lambda$ and $\nu$ are solutions to the optimization problem:*

$$\min_{\lambda,\nu \geq 0} \lambda \delta + \nu \tilde{b} + \lambda \underset{\substack{s \sim d^{\pi_{\theta_k}} \\ a \sim \pi^*}}{\mathbb{E}} \left[ \log Z_{\lambda,\nu}(s) \right] \tag{8}$$

The form of the optimal policy is intuitive, it gives high probability mass to areas of the state-action space with high return which is offset by a penalty times the cost advantage. We will refer to the optimal solution to (4-6) as the *optimal update policy*. We also note that it is possible to extend our results to accommodate for multiple constraints by introducing Lagrange multipliers $\nu_1, \ldots, \nu_m \geq 0$, one for each cost constraint and applying a similar duality argument.

Another desirable property of the optimal update policy $\pi^*$ is that for any feasible policy $\pi_{\theta_k}$, it satisfies the following bound for worst-case guarantee for cost constraint satisfaction from Achiam et al. (2017):

$$J_C(\pi^*) \leq b + \frac{\sqrt{2\delta}\gamma\epsilon_C^{\pi^*}}{(1 - \gamma)^2} \tag{9}$$

where $\epsilon_C^{\pi^*} = \max_s \left| \mathbb{E}_{a \sim \pi}[A_C^{\pi_{\theta_k}}(s, a)] \right|$.

## 3.2 Approximating the Optimal Update Policy

When solving Problem (4-6), we allow $\pi$ to be in the set of all stationary policies $\Pi$ thus the resulting $\pi^*$ is not necessarily in the parameterized policy space $\Pi_\theta$ and we may no longer be able to evaluate or sample from $\pi^*$. Therefore in the second step we project the optimal update policy back into the parameterized policy space by minimizing the loss function:

$$\mathcal{L}(\theta) = \mathop{\mathbb{E}}_{s \sim d^{\pi_{\theta_k}}} \left[ D_{\text{KL}} \left( \pi_\theta \| \pi^* \right) [s] \right] \tag{10}$$

Here $\pi_\theta \in \Pi_\theta$ is some projected policy which we will use to approximate the optimal update policy. We can use first-order methods to minimize this loss function where we make use of the following result:

**Corollary 1.** *The gradient of $\mathcal{L}(\theta)$ takes the form*

$$\nabla_\theta \mathcal{L}(\theta) = \mathop{\mathbb{E}}_{s \sim d^{\pi_{\theta_k}}} \left[ \nabla_\theta D_{KL} \left( \pi_\theta \| \pi^* \right) [s] \right] \tag{11}$$

*where*

$$\nabla_\theta D_{KL} \left( \pi_\theta \| \pi^* \right) [s] = \nabla_\theta D_{KL} \left( \pi_\theta \| \pi_{\theta_k} \right) [s] - \frac{1}{\lambda} \mathop{\mathbb{E}}_{a \sim \pi_{\theta_k}} \left[ \frac{\nabla_\theta \pi_\theta(a|s)}{\pi_{\theta_k}(a|s)} \left( A^{\pi_{\theta_k}}(s,a) - \nu A_C^{\pi_{\theta_k}}(s,a) \right) \right] \tag{12}$$

*Proof.* See Appendix B of the supplementary materials. $\qquad\square$

Note that (11) can be estimated by sampling from the trajectories generated by policy $\pi_{\theta_k}$ which allows us to train our policy using stochastic gradients.

Corollary 1 provides an outline for our algorithm. At every iteration we begin with a policy $\pi_{\theta_k}$, which we use to run trajectories and gather data. We use that data and (8) to first estimate $\lambda$ and $\nu$. We then draw a minibatch from the data to estimate $\nabla_\theta \mathcal{L}(\theta)$ given in Corollary 1. After taking a gradient step using Equation (11), we draw another minibatch and repeat the process.

## 3.3 Practical Implementation

Solving the dual problem (8) is computationally impractical for large state/action spaces as it requires calculating the partition function $Z_{\lambda,\nu}(s)$ which often involves evaluating a high-dimensional integral or sum. Furthermore, $\lambda$ and $\nu$ depends on $k$ and should be adapted at every iteration.

We note that as $\lambda \to 0$, $\pi^*$ approaches a greedy policy; as $\lambda$ increases, the policy becomes more exploratory. We also note that $\lambda$ is similar to the temperature term used in maximum entropy reinforcement learning (Ziebart et al., 2008), which has been shown to produce reasonable results when kept fixed during training (Schulman et al., 2017a; Haarnoja et al., 2018). In practice, we found that a fixed $\lambda$ found through hyperparameter sweeps provides good results. However $\nu$ needs to be continuously adapted during training so as to ensure cost constraint satisfaction. Here we appeal to an intuitive heuristic for determining $\nu$ based on primal-dual gradient methods (Bertsekas, 2014). Recall that by strong duality, the optimal $\lambda^*$ and $\nu^*$ minimizes the dual function (8) which we will denote by $L(\pi^*, \lambda, \nu)$. We can therefore apply gradient descent w.r.t. $\nu$ to minimize $L(\pi^*, \lambda, \nu)$. We can show that

**Corollary 2.** *The derivative of $L(\pi^*, \lambda, \nu)$ w.r.t. $\nu$ is*

$$\frac{\partial L(\pi^*, \lambda, \nu)}{\partial \nu} = \tilde{b} - \mathop{\mathbb{E}}_{\substack{s \sim d^{\pi_{\theta_k}} \\ a \sim \pi^*}} \left[ A^{\pi_{\theta_k}}(s,a) \right] \tag{13}$$

*Proof.* See Appendix C of the supplementary materials. $\qquad\square$

The last term in the gradient expression in Equation (13) cannot be evaluated since we do not have access to $\pi^*$. However since $\pi_{\theta_k}$ and $\pi^*$ are 'close' (by constraint (6)), it is reasonable to assume that $E_{s \sim d^{\pi_{\theta_k}}, a \sim \pi^*}[A^{\pi_{\theta_k}}(s,a)] \approx E_{s \sim d^{\pi_{\theta_k}}, a \sim \pi_{\theta_k}}[A^{\pi_{\theta_k}}(s,a)] = 0$. In practice we find that this term can be set to zero which gives the update term:

$$\nu \leftarrow \mathop{\text{proj}}_{\nu} \left[ \nu - \alpha(b - J_C(\pi_{\theta_k})) \right] \tag{14}$$

where $\alpha$ is the step size, here we have incorporated the discount term $(1-\gamma)$ in $\tilde{b}$ into the step size. The projection operator $\text{proj}_\nu$ projects $\nu$ back into the interval $[0, \nu_{\max}]$ where $\nu_{\max}$ is chosen so that $\nu$ does not become too large. However we will show in later sections that FOCOPS is generally insensitive to the choice of $\nu_{\max}$ and setting $\nu_{\max} = +\infty$ does not appear to greatly reduce performance. Practically, $J_C(\pi_{\theta_k})$ can be estimated via Monte Carlo methods using trajectories collected from $\pi_{\theta_k}$. We note that the update rule in Equation (14) is similar in to the update rule introduced in Chow et al. (2017). We recall that in (7), $\nu$ acts as a cost penalty term where increasing $\nu$ makes it less likely for state-action pairs with higher costs to be sampled by $\pi^*$. Hence in this regard, the update rule in (14) is intuitive in that it increases $\nu$ if $J_C(\pi_{\theta_k}) > b$ (i.e. the cost constraint is violated for $\pi_{\theta_k}$) and decreases $\nu$ otherwise. Using the update rule (14), we can then perform one update step on $\nu$ before updating the policy parameters $\theta$.

Our method is a first-order method, so the approximations that we make is only accurate near the initial condition (i.e. $\pi_\theta = \pi_{\theta_k}$). In order to better enforce this we also add to (11) a per-state acceptance indicator function $I(s_j) := \mathbf{1}_{D_{\text{KL}}(\pi_\theta \| \pi_{\theta_k})[s_j] \leq \delta}$. This way sampled states whose $D_{\text{KL}}(\pi_\theta \| \pi_{\theta_k})[s]$ is too large are rejected from the gradient update. The resulting sample gradient update term is

$$\hat{\nabla}_\theta \mathcal{L}(\theta) \approx \frac{1}{N} \sum_{j=1}^{N} \left[ \nabla_\theta D_{\text{KL}}(\pi_\theta \| \pi_{\theta_k})[s_j] - \frac{1}{\lambda} \frac{\nabla_\theta \pi_\theta(a_j|s_j)}{\pi_{\theta_k}(a_j|s_j)} \left( \hat{A}(s_j, a_j) - \nu \hat{A}_C(s_j, a_j) \right) \right] I(s_j).$$
(15)

Here $N$ is the number of samples we collected using policy $\pi_{\theta_k}$, $\hat{A}$ and $\hat{A}_C$ are estimates of the advantage functions (for the return and cost) obtained from critic networks. We estimate the advantage functions using the Generalized Advantage Estimator (GAE) (Schulman et al., 2016). We can then apply stochastic gradient descent using Equation (15). During training, we use the early stopping criteria $\frac{1}{N} \sum_{i=1}^{N} D_{\text{KL}}(\pi_\theta \| \pi_{\theta_k})[s_i] > \delta$ which helps prevent trust region constraint violation for the new updated policy. We update the parameters for the value net by minimizing the Mean Square Error (MSE) of the value net output and some target value (which can be estimated via Monte Carlo or bootstrap estimates of the return). We emphasize again that FOCOPS only requires first order methods (gradient descent) and is thus extremely simple to implement.

Algorithm 1 presents a summary of the FOCOPS algorithm. A more detailed pseudocode is provided in Appendix F of the supplementary materials.

---
**Algorithm 1** FOCOPS Outline
---
**Initialize:** Policy network $\pi_{\theta_0}$, Value networks $V_{\phi_0}, V_{\psi_0}^C$.
 1: **while** Stopping criteria not met **do**
 2:     Generate trajectories $\tau \sim \pi_{\theta_k}$.
 3:     Estimate $C$-returns and advantage functions.
 4:     Update $\nu$ using Equation (14).
 5:     **for** $K$ epochs **do**
 6:         **for** each minibatch **do**
 7:             Update value networks by minimizing MSE of $V_{\phi_k}, V_{\phi_k}^{\text{target}}$ and $V_{\psi_k}^C, V_{\psi_k}^{C,\text{target}}$.
 8:             Update policy network using Equation (15)
 9:         **if** $\frac{1}{N} \sum_{j=1}^{N} D_{\text{KL}}(\pi_\theta \| \pi_{\theta_k})[s_j] > \delta$ **then**
10:             Break out of inner loop
---

## 4   Experiments

We designed two different sets of experiments to test the efficacy of the FOCOPS algorithm. In the first set of experiments, we train different robotic agents to move along a straight line or a two dimensional plane, but the speed of the robot is constrained for safety purposes. The second set of experiments is inspired by the Circle experiments from Achiam et al. (2017). Both sets of experiments are implemented using the OpenAI Gym API (Brockman et al., 2016) for the MuJoCo physical simulator (Todorov et al., 2012). Implementation details for all experiments can be found in the supplementary material.

In addition to the CPO algorithm, we are also including for comparison two algorithms based on Lagrangian methods (Bertsekas, 1997), which uses adaptive penalty coefficients to enforce constraints. For an objective function $f(\theta)$ and constraint $g(\theta) \leq 0$, the Lagrangian method solves max-min optimization problem $\max_\theta \min_{\nu \geq 0}(f(\theta) - \nu g(\theta))$. These methods first perform gradient ascent on $\theta$, and then gradient descent on $\nu$. Chow et al. (2019) and Ray et al. (2019) combined Lagrangian method with PPO (Schulman et al., 2017b) and TRPO (Schulman et al., 2015) to form the PPO Lagrangian and TRPO Lagrangian algorithms, which we will subsequently abbreviate as PPO-L and TRPO-L respectively. Details for these two algorithms can be found in the supplementary material.

## 4.1 Robots with Speed Limit

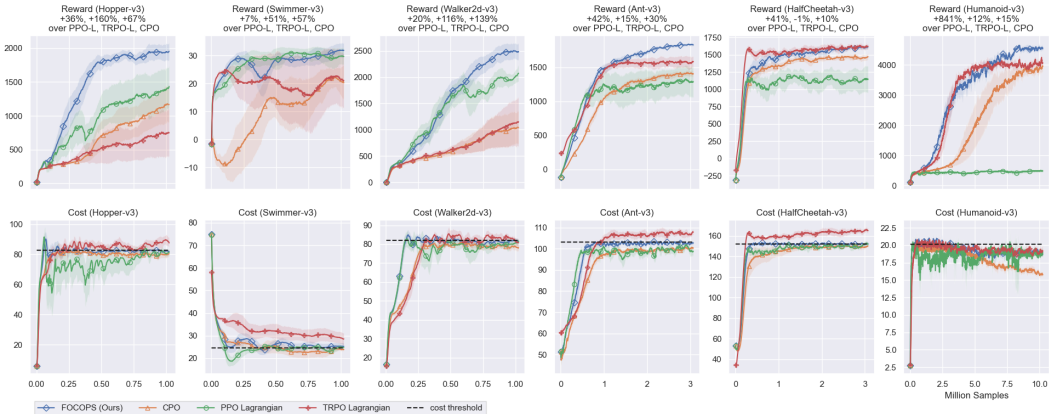

Figure 1: Learning curves for robots with speed limit tasks. The $x$-axis represent the number of samples used and the $y$-axis represent the average total reward/cost return of the last 100 episodes. The solid line represent the mean of 1000 bootstrap samples over 10 random seeds. The shaded regions represent the bootstrap normal $95\%$ confidence interval. FOCOPS consistently enforce approximate constraint satisfaction while having a higher performance on five out of the six tasks.

We consider six MuJoCo environments where we attempt to train a robotic agent to walk. However we impose a speed limit on our environments. The cost thresholds are calculated using 50% of the speed attained by an unconstrained PPO agent after training for a million samples (Details can be found in Appendix G.1).

Figure 1 shows that FOCOPS outperforms other baselines in terms of reward on most tasks while enforcing the cost constraint. In theory, FOCOPS assumes that the initial policy is feasible. This assumption is violated in the Swimmer-v3 environment. However in practice, the gradient update term increases the dual variable associated with the cost when the cost constraint is violated, this would result in a feasible policy after a certain number of iterations. We observed that this is indeed the case with the swimmer environment (and similarly the AntCircle environment in the next section). Note also that Lagrangian methods outperform CPO on several environments in terms of reward, this is consistent with the observation made by Ray et al. (2019) and Stooke et al. (2020). However on most tasks TRPO-L does not appear to consistently maintain constraint satisfaction during training. For example on HalfCheetah-v3, even though TRPO-L outperforms FOCOPS in terms of total return, it violates the cost constraint by nearly 9%. PPO-L is shown to do well on simpler tasks but performance deteriorates drastically on the more challenging environments (Ant-v3, HalfCheetah-v3, and Humanoid-v3), this is in contrast to FOCOPS which perform particularly well on these set of tasks. In Table 1 we summarized the performance of all four algorithms.

## 4.2 Circle Tasks

For these tasks, we use the same exact geometric setting, reward, and cost constraint function as Achiam et al. (2017), a geometric illustration of the task and details on the reward/cost functions can

Table 1: Bootstrap mean and normal 95% confidence interval with 1000 bootstrap samples over 10 random seeds of reward/cost return after training on robot with speed limit environments. Cost thresholds are in brackets under the environment names.

| Environment | | PPO-L | TRPO-L | CPO | **FOCOPS** |
|---|---|---|---|---|---|
| Ant-v3 | Reward | $1291.4 \pm 216.4$ | $1585.7 \pm 77.5$ | $1406.0 \pm 46.6$ | $\mathbf{1830.0 \pm 22.6}$ |
| (103.12) | Cost | $98.78 \pm 1.77$ | $107.82 \pm 1.16$ | $100.25 \pm 0.67$ | $102.75 \pm 1.08$ |
| HalfCheetah-v3 | Reward | $1141.3 \pm 192.4$ | $\mathbf{1621.59 \pm 39.4}$ | $1470.8 \pm 40.0$ | $1612.2 \pm 25.9$ |
| (151.99) | Cost | $151.53 \pm 1.88$ | $164.93 \pm 2.43$ | $150.05 \pm 1.40$ | $152.36 \pm 1.55$ |
| Hopper-v3 | Reward | $1433.8 \pm 313.3$ | $750.3 \pm 355.3$ | $1167.1 \pm 257.6$ | $\mathbf{1953.4 \pm 127.3}$ |
| (82.75) | Cost | $81.29 \pm 2.34$ | $87.57 \pm 3.48$ | $80.39 \pm 1.39$ | $81.84 \pm 0.92$ |
| Humanoid-v3 | Reward | $471.3 \pm 49.0$ | $4062.4 \pm 113.3$ | $3952.7 \pm 174.4$ | $\mathbf{4529.7 \pm 86.2}$ |
| (20.14) | Cost | $18.89 \pm 0.77$ | $19.23 \pm 0.76$ | $15.83 \pm 0.41$ | $18.63 \pm 0.37$ |
| Swimmer-v3 | Reward | $29.73 \pm 3.13$ | $21.15 \pm 9.56$ | $20.31 \pm 6.01$ | $\mathbf{31.94 \pm 2.60}$ |
| (24.52) | Cost | $24.72 \pm 0.85$ | $28.57 \pm 2.68$ | $23.88 \pm 0.64$ | $25.29 \pm 1.49$ |
| Walker2d-v3 | Reward | $2074.4 \pm 155.7$ | $1153.1 \pm 473.3$ | $1040.0 \pm 303.3$ | $\mathbf{2485.9 \pm 158.3}$ |
| (81.89) | Cost | $81.7 \pm 1.14$ | $80.79 \pm 2.13$ | $78.12 \pm 1.78$ | $81.27 \pm 1.33$ |

be found in Appendix G.2 of the supplementary materials. The goal of the agents is to move along the circumference of a circle while remaining within a safe region smaller than the radius of the circle.

Similar to the previous tasks, we provide learning curves (Figure 2) and numerical summaries (Table 2) of the experiments. We also plotted an unconstrained PPO agent for comparison. On these tasks, all four approaches are able to approximately enforce cost constraint satisfaction (set at 50), but FOCOPS does so while having a higher performance. Note for both tasks, the 95% confidence interval for FOCOPS lies above the confidence intervals for all other algorithms, this is strong indication that FOCOPS outperforms the other three algorithms on these particular tasks.

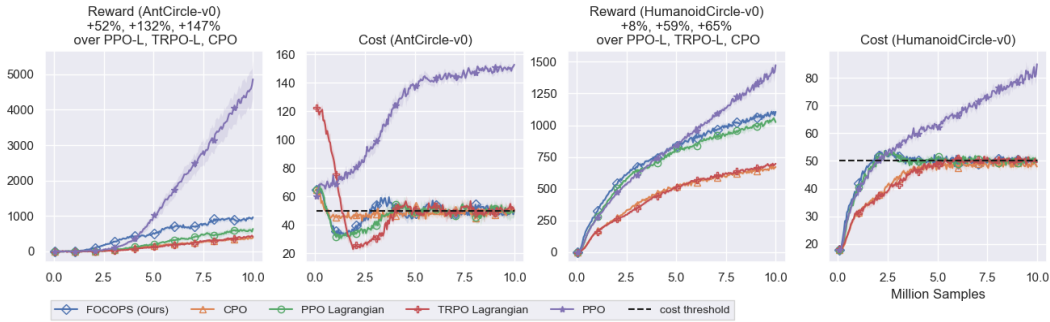

Figure 2: Comparing reward and cost returns on circle Tasks. The $x$-axis represent the number of samples used and the $y$-axis represent the average total reward/cost return of the last 100 episodes. The solid line represent the mean of 1000 bootstrap samples over 10 random seeds. The shaded regions represent the bootstrap normal 95% confidence interval. An unconstrained PPO agent is also plotted for comparison.

## 4.3 Generalization Analysis

In supervised learning, the standard approach is to use separate datasets for training, validation, and testing where we can then use some evaluation metric on the test set to determine how well an algorithm generalizes over unseen data. However such a scheme is not suitable for reinforcement learning.

Table 2: Bootstrap mean and normal 95% confidence interval with 1000 bootstrap samples over 10 random seeds of reward/cost return after training on circle environments for 10 million samples. Cost thresholds are in brackets under the environment names.

| Environment | | PPO-L | TRPO-L | CPO | **FOCOPS** |
|---|---|---|---|---|---|
| Ant-Circle | Reward | $637.4 \pm 88.2$ | $416.7 \pm 42.1$ | $390.9 \pm 43.9$ | $\mathbf{965.9 \pm 46.2}$ |
| (50.0) | Cost | $50.4 \pm 4.4$ | $50.4 \pm 3.9$ | $50.0 \pm 3.5$ | $49.9 \pm 2.2$ |
| Humanoid-Circle | Reward | $1024.5 \pm 23.4$ | $697.5 \pm 14.0$ | $671.0 \pm 12.5$ | $\mathbf{1106.1 \pm 32.2}$ |
| (50.0) | Cost | $50.3 \pm 0.59$ | $49.6 \pm 0.96$ | $47.9 \pm 1.5$ | $49.9 \pm 0.8$ |

To similarly evaluate our reinforcement learning agent, we first trained our agent on a fixed random seed. We then tested the trained agent on ten unseen random seeds (Pineau, 2018). We found that with the exception of Hopper-v3, FOCOPS outperformed every other constrained algorithm on all robots with speed limit environments. Detailed analysis of the generalization results are provided in Appendix H of the supplementary materials.

## 4.4 Sensitivity Analysis

The Lagrange multipliers play a key role in the FOCOPS algorithms, in this section we explore the sensitivity of the hyperparameters $\lambda$ and $\nu_{\max}$. We find that the performance of FOCOPS is largely insensitive to the choice of these hyperparamters. To demonstrate this, we conducted a series of experiments on the robots with speed limit tasks.

The hyperparameter $\nu_{\max}$ was selected via hyperparameter sweep on the set $\{1, 2, 3, 5, 10, +\infty\}$. However we found that FOCOPS is not sensitive to the choice of $\nu_{\max}$ where setting $\nu_{\max} = +\infty$ only leads to an average $0.3\%$ degradation in performance compared to the optimal $\nu_{\max} = 2$. Similarly we tested the performance of FOCOPS against different values of $\lambda$ where for other values of $\lambda$, FOCOPS performs on average $7.4\%$ worse compared to the optimal $\lambda = 1.5$. See Appendix I of the supplementary materials for more details.

## 5 Discussion

We introduced FOCOPS—a simple first-order approach for training RL agents with safety constraints. FOCOPS is theoretically motivated and is shown to empirically outperform more complex second-order methods. FOCOPS is also easy to implement. We believe in the value of simplicity as it makes RL more accessible to researchers in other fields who wish to apply such methods in their own work. Our results indicate that constrained RL is an effective approach for addressing RL safety and can be efficiently solved using our two step approach.

There are a number of promising avenues for future work: such as incorporating off-policy data; studying how our two-step approach can deal with different types of constraints such as sample path constraints (Ross and Varadarajan, 1989, 1991), or safety constraints expressed in other more natural forms such as human preferences (Christiano et al., 2017) or natural language (Luketina et al., 2019).

## 6 Broader Impact

Safety is a critical element in real-world applications of RL. We argue in this paper that scalar reward signals alone is often insufficient in motivating the agent to avoid harmful behavior. An RL systems designer needs to carefully balance how much to encourage desirable behavior and how much to penalize unsafe behavior where too much penalty could prevent the agent from sufficient exploration and too little could lead to hazardous consequences. This could be extremely difficult in practice. Constraints are a more natural way of quantifying safety requirements and we advocate for researchers to consider including constraints in safety-critical RL systems.

## Acknowledgments and Disclosure of Funding

This research was not supported by any external funding. The authors would like to express our appreciation for the technical support provided by the NYU Shanghai High Performance Computing (HPC) administrator Zhiguo Qi and the HPC team at NYU. We would also like to thank Joshua Achiam for his insightful comments on an earlier draft of this paper and for making his implementation of the CPO algorithm publicly available.

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
