[Supplementary Material 1 · focops_full_paper.pdf]

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

# Supplementary Material for First Order Constrained Optimization in Policy Space

# Appendices

## A   Proof of Theorem 1

**Theorem 1.** *Let* $\tilde{b} = (1-\gamma)(b - \tilde{J}_C(\pi_{\theta_k}))$. *If* $\pi_{\theta_k}$ *is a feasible solution, the optimal policy for (4-6) takes the form*

$$\pi^*(a|s) = \frac{\pi_{\theta_k}(a|s)}{Z_{\lambda,\nu}(s)} \exp\left(\frac{1}{\lambda}\left(A^{\pi_{\theta_k}}(s,a) - \nu A_C^{\pi_{\theta_k}}(s,a)\right)\right) \tag{7}$$

*where* $Z_{\lambda,\nu}(s)$ *is the partition function which ensures (7) is a valid probability distribution,* $\lambda$ *and* $\nu$ *are solutions to the optimization problem:*

$$\min_{\lambda,\nu \geq 0}\ \lambda\delta + \nu\tilde{b} + \lambda \mathop{\mathbb{E}}_{\substack{s \sim d^{\pi_{\theta_k}} \\ a \sim \pi^*}}[\log Z_{\lambda,\nu}(s)] \tag{8}$$

*Proof.* We will begin by showing that Problem (4-6) is convex w.r.t. $\pi = \{\pi(a|s) : s \in \mathcal{S}, a \in \mathcal{A}\}$. First note that the objective function is linear w.r.t. $\pi$. Since $J_C(\pi_{\theta_k})$ is a constant w.r.t. $\pi$, constraint (5) is linear. Constraint (6) can be rewritten as $\sum_s d^{\pi_{\theta_k}}(s)D_{\mathrm{KL}}(\pi\|\pi_{\theta_k})[s] \leq \delta$, the KL divergence is convex w.r.t. its first argument, therefore constraint (6) which is a linear combination of convex functions is also convex. Since $\pi_{\theta_k}$ satisfies Constraint (5) and is also an interior point within the set given by Constraints (6) ($D_{\mathrm{KL}}(\pi_{\theta_k}\|\pi_{\theta_k}) = 0$, and $\delta > 0$), therefore Slater's constraint qualification holds, strong duality holds.

We can therefore solve for the optimal value of Problem (4-6) $p^*$ by solving the corresponding dual problem. Let

$$L(\pi,\lambda,\nu) = \lambda\delta + \nu\tilde{b} + \mathop{\mathbb{E}}_{s \sim d^{\pi_{\theta_k}}}\left[\mathop{\mathbb{E}}_{a \sim \pi(\cdot|s)}[A^{\pi_{\theta_k}}(s,a)] - \nu \mathop{\mathbb{E}}_{a \sim \pi(\cdot|s)}[A_C^{\pi_{\theta_k}}(s,a)] - \lambda D_{\mathrm{KL}}(\pi\|\pi_{\theta_k})[s]\right] \tag{16}$$

Therefore,

$$p^* = \max_{\pi \in \Pi}\min_{\lambda,\nu \geq 0} L(\pi,\lambda,\nu) = \min_{\lambda,\nu \geq 0}\max_{\pi \in \Pi} L(\pi,\lambda,\nu) \tag{17}$$

where we invoked strong duality in the second equality. We note that if $\pi^*, \lambda^*, \nu^*$ are optimal for (17), $\pi^*$ is also optimal for Problem (4-6) (Boyd and Vandenberghe, 2004).

Consider the inner maximization problem in (17), we can decompose this problem into separate problems, one for each $s$. This gives us an optimization problem of the form,

$$\begin{aligned} \underset{\pi}{\text{maximize}} \quad & \mathop{\mathbb{E}}_{a \sim \pi(\cdot|s)}\left[A^{\pi_{\theta_k}}(s,a) - \nu A_C^{\pi_{\theta_k}}(s,a) - \lambda(\log\pi(a|s) - \log\pi_{\theta_k}(a|s))\right] \\ \text{subject to} \quad & \sum_a \pi(a|s) = 1 \\ & \pi(a|s) \geq 0 \quad \text{for all } a \in \mathcal{A} \end{aligned} \tag{18}$$

which is equivalent to the inner maximization problem in (17). This is clearly a convex optimization problem which we can solve using a simple Lagrangian argument. We can write the Lagrangian of (18) as

$$G(\pi) = \sum_a \pi(a|s)\left[A^{\pi_{\theta_k}}(s,a) - \nu A_C^{\pi_{\theta_k}}(s,a) - \lambda(\log\pi(a|s) - \log\pi_{\theta_k}(a|s)) + \zeta\right] - 1 \tag{19}$$

where $\zeta > 0$ is the Lagrange multiplier associated with the constraint $\sum_a \pi(a|s) = 1$. Differentiating $G(\pi)$ w.r.t. $\pi(a|s)$ for some $a$:

$$\frac{\partial G}{\partial \pi(a|s)} = A^{\pi_{\theta_k}}(s,a) - \nu A_C^{\pi_{\theta_k}}(s,a) - \lambda(\log \pi(a|s) + 1 - \log \pi_{\theta_k}(a|s)) + \zeta \qquad (20)$$

Setting (20) to zero and rearranging the term, we obtain

$$\pi(a|s) = \pi_{\theta_k}(a|s) \exp\left(\frac{1}{\lambda}\left(A^{\pi_{\theta_k}}(s,a) - \nu A_C^{\pi_{\theta_k}}(s,a)\right) + \frac{\zeta}{\lambda} + 1\right) \qquad (21)$$

We chose $\zeta$ so that $\sum_a \pi(a|s) = 1$ and rewrite $\zeta/\lambda + 1$ as $Z_{\lambda,\nu}(s)$. We find that the optimal solution $\pi^*$ to (18) takes the form

$$\pi^*(a|s) = \frac{\pi_{\theta_k}(a|s)}{Z_{\lambda,\nu}(s)} \exp\left(\frac{1}{\lambda}\left(A^{\pi_{\theta_k}}(s,a) - \nu A_C^{\pi_{\theta_k}}(s,a)\right)\right)$$

Plugging $\pi^*$ back into Equation 17 gives us

$$\begin{aligned}
p^* &= \min_{\lambda,\nu \geq 0} \lambda\delta + \nu\tilde{b} + \mathop{\mathbb{E}}_{\substack{s \sim d^{\pi_{\theta_k}} \\ a \sim \pi^*}} [A^{\pi_{\theta_k}}(s,a) - \nu A_C^{\pi_{\theta_k}}(s,a) - \lambda(\log \pi^*(a|s) - \log \pi_{\theta_k}(a|s))] \\
&= \min_{\lambda,\nu \geq 0} \lambda\delta + \nu\tilde{b} + \mathop{\mathbb{E}}_{\substack{s \sim d^{\pi_{\theta_k}} \\ a \sim \pi^*}} [A^{\pi_{\theta_k}}(s,a) - \nu A_C^{\pi_{\theta_k}}(s,a) - \lambda(\log \pi_{\theta_k}(a|s) - \log Z_{\lambda,\nu}(s) \\
&+ \frac{1}{\lambda}(A^{\pi_{\theta_k}}(s,a) - \nu A_C^{\pi_{\theta_k}}(s,a)) - \log \pi_{\theta_k}(a|s))] \\
&= \min_{\lambda,\nu \geq 0} \lambda\delta + \nu\tilde{b} + \lambda \mathop{\mathbb{E}}_{\substack{s \sim d^{\pi_{\theta_k}} \\ a \sim \pi^*}} [\log Z_{\lambda,\nu}(s)]
\end{aligned}$$

$\square$

# B    Proof of Corollary 1

**Corollary 1.** *The gradient of $\mathcal{L}(\theta)$ takes the form*

$$\nabla_\theta \mathcal{L}(\theta) = \mathop{\mathbb{E}}_{s \sim d^{\pi_{\theta_k}}} [\nabla_\theta D_{KL}(\pi_\theta \| \pi^*)[s]] \qquad (11)$$

*where*

$$\nabla_\theta D_{KL}(\pi_\theta \| \pi^*)[s] = \nabla_\theta D_{KL}(\pi_\theta \| \pi_{\theta_k})[s] - \frac{1}{\lambda} \mathop{\mathbb{E}}_{a \sim \pi_{\theta_k}} \left[\frac{\nabla_\theta \pi_\theta(a|s)}{\pi_{\theta_k}(a|s)}\left(A^{\pi_{\theta_k}}(s,a) - \nu A_C^{\pi_{\theta_k}}(s,a)\right)\right] \quad (12)$$

*Proof.* We only need to calculate the gradient of the loss function for a single sampled $s$. We first note that,

$$D_{KL}(\pi_\theta \| \pi^*)[s] = -\sum_a \pi_\theta(a|s) \log \pi^*(a|s) + \sum_a \pi_\theta(a|s) \log \pi_\theta(a|s)$$

$$= H(\pi_\theta, \pi^*)[s] - H(\pi_\theta)[s]$$

where $H(\pi_\theta)[s]$ is the entropy and $H(\pi_\theta, \pi^*)[s]$ is the cross-entropy under state $s$. We expand the cross entropy term which gives us

$$\begin{aligned}
H(\pi_\theta, \pi^*)[s] &= -\sum_a \pi_\theta(a|s) \log \pi^*(a|s) \\
&= -\sum_a \pi_\theta(a|s) \log\left(\frac{\pi_{\theta_k}(a|s)}{Z_{\lambda,\nu}(s)} \exp\left[\frac{1}{\lambda}\left(A^{\pi_{\theta_k}}(s,a) - \nu A_C^{\pi_{\theta_k}}(s,a)\right)\right]\right) \\
&= -\sum_a \pi_\theta(a|s) \log \pi_{\theta_k}(a|s) + \log Z_{\lambda,\nu}(s) - \frac{1}{\lambda} \sum_a \pi_\theta(a|s)\left(A^{\pi_{\theta_k}}(s,a) - \nu A_C^{\pi_{\theta_k}}(s,a)\right)
\end{aligned}$$

We then subtract the entropy term to recover the KL divergence:

$$D_{\mathrm{KL}}\left(\pi_\theta \| \pi^*\right)[s] = D_{\mathrm{KL}}\left(\pi_\theta \| \pi_{\theta_k}\right)[s] + \log Z_{\lambda,\nu}(s) - \frac{1}{\lambda} \sum_a \pi_\theta(a|s)\left(A^{\pi_{\theta_k}}(s,a) - \nu A_C^{\pi_{\theta_k}}(s,a)\right)$$

$$= D_{\mathrm{KL}}\left(\pi_\theta \| \pi_{\theta_k}\right)[s] + \log Z_{\lambda,\nu}(s) - \frac{1}{\lambda} \mathop{\mathbb{E}}_{a\sim\pi_{\theta_k}(\cdot|s)}\left[\frac{\pi_\theta(a|s)}{\pi_{\theta_k}(a|s)}\left(A^{\pi_{\theta_k}}(s,a) - \nu A_C^{\pi_{\theta_k}}(s,a)\right)\right]$$

where in the last equality we applied importance sampling to rewrite the expectation w.r.t. $\pi_{\theta_k}$. Finally, taking the gradient on both sides gives us:

$$\nabla_\theta D_{\mathrm{KL}}\left(\pi_\theta \| \pi^*\right)[s] = \nabla_\theta D_{\mathrm{KL}}\left(\pi_\theta \| \pi_{\theta_k}\right)[s] - \frac{1}{\lambda} \mathop{\mathbb{E}}_{a\sim\pi_{\theta_k}(\cdot|s)}\left[\frac{\nabla_\theta \pi_\theta(a|s)}{\pi_{\theta_k}(a|s)}\left(A^{\pi_{\theta_k}}(s,a) - \nu A_C^{\pi_{\theta_k}}(s,a)\right)\right].$$

$\square$

## C  Proof of Corollary 2

**Corollary 2.** *The derivative of $L(\pi^*, \lambda, \nu)$ w.r.t. $\nu$ is*

$$\frac{\partial L(\pi^*, \lambda, \nu)}{\partial \nu} = \tilde{b} - \mathop{\mathbb{E}}_{\substack{s\sim d^{\pi_{\theta_k}} \\ a\sim\pi^*}}\left[A^{\pi_{\theta_k}}(s,a)\right] \tag{13}$$

*Proof.* From Theorem 1, we have

$$L(\pi^*, \lambda, \nu) = \lambda\delta + \nu\tilde{b} + \lambda \mathop{\mathbb{E}}_{\substack{s\sim d^{\pi_{\theta_k}} \\ a\sim\pi^*}}\left[\log Z_{\lambda,\nu}(s)\right]. \tag{22}$$

The first two terms is an affine function w.r.t. $\nu$, therefore its derivative is $\tilde{b}$. We will then focus on the expectation in the last term. To simplify our derivation, we will first calculate the derivative of $\pi^*$ w.r.t. $\nu$,

$$\frac{\partial\pi^*(a|s)}{\partial\nu} = \frac{\pi_{\theta_k}(a|s)}{Z_{\lambda,\nu}^2(s)}\left[Z_{\lambda,\nu}(s)\frac{\partial}{\partial\nu}\exp\left(\frac{1}{\lambda}\left(A^{\pi_{\theta_k}}(s,a) - \nu A_C^{\pi_{\theta_k}}(s,a)\right)\right)\right.$$
$$\left. - \exp\left(\frac{1}{\lambda}\left(A^{\pi_{\theta_k}}(s,a) - \nu A_C^{\pi_{\theta_k}}(s,a)\right)\right)\frac{\partial Z_{\lambda,\nu}(s)}{\partial\nu}\right]$$
$$= -\frac{A_C^{\pi_{\theta_k}}(s,a)}{\lambda}\pi^*(a|s) - \pi^*(a|s)\frac{\partial\log Z_{\lambda,\nu}(s)}{\partial\nu}$$

Therefore the derivative of the expectation in the last term of $L(\pi^*, \lambda, \nu)$ can be written as

$$\frac{\partial}{\partial\nu} \mathop{\mathbb{E}}_{\substack{s\sim d^{\pi_{\theta_k}} \\ a\sim\pi^*}}\left[\log Z_{\lambda,\nu}(s)\right]$$

$$= \mathop{\mathbb{E}}_{\substack{s\sim d^{\pi_{\theta_k}} \\ a\sim\pi_{\theta_k}}}\left[\frac{\partial}{\partial\nu}\left(\frac{\pi^*(a|s)}{\pi_{\theta_k}(a|s)}\log Z_{\lambda,\nu}(s)\right)\right]$$

$$= \mathop{\mathbb{E}}_{\substack{s\sim d^{\pi_{\theta_k}} \\ a\sim\pi_{\theta_k}}}\left[\frac{1}{\pi_{\theta_k}(a|s)}\left(\frac{\partial\pi^*(a|s)}{\partial\nu}\log Z_{\lambda,\nu}(s) + \pi^*(a|s)\frac{\partial\log Z_{\lambda,\nu}(s)}{\partial\nu}\right)\right]$$

$$= \mathop{\mathbb{E}}_{\substack{s\sim d^{\pi_{\theta_k}} \\ a\sim\pi_{\theta_k}}}\left[\frac{\pi^*(a|s)}{\pi_{\theta_k}(a|s)}\left(-\frac{A_C^{\pi_{\theta_k}}(s,a)}{\lambda}\log Z_{\lambda,\nu}(s) - \frac{\partial\log Z_{\lambda,\nu}(s)}{\partial\nu}\log Z_{\lambda,\nu}(s) + \frac{\partial\log Z_{\lambda,\nu}(s)}{\partial\nu}\right)\right]$$

$$= \mathop{\mathbb{E}}_{\substack{s\sim d^{\pi_{\theta_k}} \\ a\sim\pi^*}}\left[-\frac{A_C^{\pi_{\theta_k}}(s,a)}{\lambda}\log Z_{\lambda,\nu}(s) - \frac{\partial\log Z_{\lambda,\nu}(s)}{\partial\nu}\log Z_{\lambda,\nu}(s) + \frac{\partial\log Z_{\lambda,\nu}(s)}{\partial\nu}\right].$$

$$\tag{23}$$

Also,

$$
\begin{aligned}
\frac{\partial Z_{\lambda,\nu}(s)}{\partial \nu} &= \frac{\partial}{\partial \nu} \sum_a \pi_{\theta_k}(a|s) \exp\left(\frac{1}{\lambda}\left(A^{\pi_{\theta_k}}(s,a) - \nu A_C^{\pi_{\theta_k}}(s,a)\right)\right) \\
&= \sum_a -\pi_{\theta_k}(a|s) \frac{A_C^{\pi_{\theta_k}}(s,a)}{\lambda} \exp\left(\frac{1}{\lambda}\left(A^{\pi_{\theta_k}}(s,a) - \nu A_C^{\pi_{\theta_k}}(s,a)\right)\right) \\
&= \sum_a -\frac{A_C^{\pi_{\theta_k}}(s,a)}{\lambda} \frac{\pi_{\theta_k}(a|s)}{Z_{\lambda,\nu}(s)} \exp\left(\frac{1}{\lambda}\left(A^{\pi_{\theta_k}}(s,a) - \nu A_C^{\pi_{\theta_k}}(s,a)\right)\right) Z_{\lambda,\nu}(s) \\
&= -\frac{Z_{\lambda,\nu}(s)}{\lambda} \mathop{\mathbb{E}}_{a\sim\pi^*(\cdot|s)}\left[A_C^{\pi_{\theta_k}}(s,a)\right].
\end{aligned}
\tag{24}
$$

Therefore,

$$
\frac{\partial \log Z_{\lambda,\nu}(s)}{\partial \nu} = \frac{\partial Z_{\lambda,\nu}(s)}{\partial \nu} \frac{1}{Z_{\lambda,\nu}(s)} = -\frac{1}{\lambda} \mathop{\mathbb{E}}_{a\sim\pi^*(\cdot|s)}\left[A_C^{\pi_{\theta_k}}(s,a)\right].
\tag{25}
$$

Plugging (25) into the last equality in (23) gives us

$$
\begin{aligned}
\frac{\partial}{\partial \nu} \mathop{\mathbb{E}}_{\substack{s\sim d^{\pi_{\theta_k}}\\ a\sim\pi^*}}[\log Z_{\lambda,\nu}(s)] &= \mathop{\mathbb{E}}_{\substack{s\sim d^{\pi_{\theta_k}}\\ a\sim\pi^*}}\left[-\frac{A_C^{\pi_{\theta_k}}(s,a)}{\lambda}\log Z_{\lambda,\nu}(s) + \frac{A_C^{\pi_{\theta_k}}(s,a)}{\lambda}\log Z_{\lambda,\nu}(s) - \frac{1}{\lambda}A_C^{\pi_{\theta_k}}(s,a)\right] \\
&= -\frac{1}{\lambda} \mathop{\mathbb{E}}_{\substack{s\sim d^{\pi_{\theta_k}}\\ a\sim\pi^*}}\left[A_C^{\pi_{\theta_k}}(s,a)\right].
\end{aligned}
\tag{26}
$$

Combining (26) with the derivatives of the affine term gives us the final desired result.

$\square$

# D  PPO Lagrangian and TRPO Lagrangian

## D.1  PPO-Lagrangian

Recall that the PPO (clipped) objective takes the form (Schulman et al., 2017b)

$$
L(\theta) = \min\left(\frac{\pi_\theta(a|s)}{\pi_{\theta_k}(a|s)}A^{\pi_{\theta_k}}(s,a), \mathrm{clip}\left(\frac{\pi_\theta(a|s)}{\pi_{\theta_k}(a|s)}, 1-\epsilon, 1+\epsilon\right) A^{\pi_{\theta_k}}(s,a)\right)
\tag{27}
$$

We augment this objective with an additional term to form the a new objective function

$$
\tilde{L}(\theta) = L(\theta) + \nu\left(J_C(\pi_\theta) - b\right)
\tag{28}
$$

The Lagrangian method involves a maximization and a minimization step. For the maximization step, we optimize the objective (28) by performing backpropagation w.r.t. $\theta$. For the minimization step, we apply gradient descent to the same objective w.r.t $\nu$. Like FOCOPS, the PPO-Lagrangian algorithm is also first-order thus simple to implement, however its empirical performance deteriorates drastically on more challenging environments (See Section 4). Furthermore, it remains an open question whether PPO-Lagrangian satisfies any worst-case constraint guarantees.

## D.2  TRPO-Lagrangian

Similar to the PPO-Lagrangian method, we instead optimize an augmented TRPO problem

$$
\underset{\theta}{\text{maximize}} \quad \tilde{L}(\theta)
\tag{29}
$$

$$
\text{subject to} \quad \bar{D}_{\mathrm{KL}}(\pi_\theta \,\|\, \pi_{\theta_k})[s] \leq \delta.
\tag{30}
$$

where

$$
\tilde{L}(\theta) = \frac{\pi_\theta(a|s)}{\pi_{\theta_k}(a|s)}A^{\pi_{\theta_k}}(s,a) + \nu\left(J_C(\pi_\theta) - b\right)
\tag{31}
$$

We then apply Taylor approximation to (29) and (30) which gives us

$$\underset{\theta}{\text{maximize}} \quad \tilde{g}^T(\theta - \theta_k) \tag{32}$$

$$\text{subject to} \quad \frac{1}{2}(\theta - \theta_k)^T H(\theta - \theta_k) \le \delta. \tag{33}$$

Here $\tilde{g}$ is the gradient of (29) w.r.t. the parameter $\theta$.

Like for the PPO-Lagrangian method, we first perform a maximization step where we optimize (32)-(33) using the TRPO method (Schulman et al., 2015). We then perform a minimization step by updating $\nu$. TRPO-Lagrangian is also a second-order algorithm. Performance-wise TRPO-Lagrangian does poorly in terms of constraint satisfaction. Like PPO-Lagrangian, further research into the theoretical properties of TRPO merits further research.

# E  FOCOPS for Different Cost Thresholds

In this section, we verify that FOCOPS works effectively for different threshold levels. We experiment on the robots with speed limits environments. For each environment, we calculated the cost required for an unconstrained PPO agent after training for 1 million samples. We then used $25\%$, $50\%$, and $75\%$ of this cost as our cost thresholds and trained FOCOPS on each of thresholds respectively. The learning curves are reported in Figure 3. We note from these plots that FOCOPS can effectively learn constraint-satisfying policies for different cost thresholds.

Figure 3: Performance of FOCOPS on robots with speed limit tasks with different cost thresholds. The $x$-axis represent the number of samples used and the $y$-axis represent the average total reward/cost return of the last 100 episodes. The solid line represent the mean of 1000 bootstrap samples over 10 random seeds. The horizontal lines in the cost plots represent the cost thresholds corresponding to $25\%$, $50\%$, and $75\%$ of the cost required by an unconstrained PPO agent trained with 1 million samples. Each solid line represents FOCOPS trained with the corresponding thresholds. The shaded regions represent the bootstrap normal $95\%$ confidence interval. Each of the solid lines represent

# F  Pseudocode

---

**Algorithm 2** First Order Constrained Optimization in Policy Space (FOCOPS)

---

**Initialize:** Policy network $\pi_\theta$; Value network for return $V_\phi$; Value network for costs $V_\psi^C$.

**Initialize:** Discount rates $\gamma$, GAE parameter $\beta$; Learning rates $\alpha_\nu, \alpha_V, \alpha_\pi$; Temperature $\lambda$; Initial cost constraint parameter $\nu$; Cost constraint parameter bound $\nu_{\max}$. Trust region bound $\delta$; Cost bound $b$.

**while** Stopping criteria not met **do**

Generate batch data of $M$ episodes of length $T$ from $(s_{i,t}, a_{i,t}, r_{i,t}, s_{i,t+1}, c_{i,t})$ from $\pi_\theta$, $i = 1, \ldots, M, t = 1, \ldots, T$.

Estimate $C$-return by averaging over $C$-return for all episodes:

$$\hat{J}_C = \frac{1}{M} \sum_{i=1}^{M} \sum_{t=0}^{T-1} \gamma^t c_{i,t}$$

Store old policy $\theta' \leftarrow \theta$

Estimate advantage functions $\hat{A}_{i,t}$ and $\hat{A}_{i,t}^C$, $i = 1, \ldots, M, t = 1, \ldots, T$ using GAE.

Get $V_{i,t}^{\text{target}} = \hat{A}_{i,t} + V_\phi(s_{i,t})$ and $V_{i,t}^{C,\text{target}} = \hat{A}_{i,t} + V_\psi^C(s_{i,t})$

Update $\nu$ by

$$\nu \leftarrow \underset{\nu}{\text{proj}} \left[ \nu - \alpha_\nu \left( b - \hat{J}_C \right) \right]$$

**for** $K$ epochs **do**

**for** each minibatch $\{s_j, a_j, A_j, A_j^C, V_j^{\text{target}}, V_j^{C,\text{target}}\}$ of size $B$ **do**

Value loss functions

$$\mathcal{L}_V(\phi) = \frac{1}{2N} \sum_{j=1}^{B} (V_\phi(s_j) - V_j^{\text{target}})^2$$

$$\mathcal{L}_{V^C}(\psi) = \frac{1}{2N} \sum_{j=1}^{B} (V_\psi(s_j) - V_j^{C,\text{target}})^2$$

Update value networks

$$\phi \leftarrow \phi - \alpha_V \nabla_\phi \mathcal{L}_V(\phi)$$
$$\psi \leftarrow \psi - \alpha_V \nabla_\psi \mathcal{L}_{V^C}(\psi)$$

Update policy

$$\theta \leftarrow \theta - \alpha_\pi \hat{\nabla}_\theta \mathcal{L}_\pi(\theta)$$

where

$$\hat{\nabla}_\theta \mathcal{L}_\pi(\theta) \approx \frac{1}{B} \sum_{j=1}^{B} \left[ \nabla_\theta D_{\text{KL}} \left( \pi_\theta \| \pi_{\theta'} \right) [s_j] - \frac{1}{\lambda} \frac{\nabla_\theta \pi_\theta(a_j|s_j)}{\pi_{\theta'}(a_j|s_j)} \left( \hat{A}_j - \nu \hat{A}_j^C \right) \right] \mathbf{1}_{D_{\text{KL}}(\pi_\theta \| \pi_{\theta'})[s_j] \leq \delta}$$

**if** $\frac{1}{MT} \sum_{i=1}^{M} \sum_{t=0}^{T-1} D_{\text{KL}} \left( \pi_\theta \| \pi_{\theta'} \right) [s_{i,t}] > \delta$ **then**
Break out of inner loop

---

# G  Implementation Details for Experiments

Our open-source implementation of FOCOPS can be found at `https://github.com/ymzhang01/focops`. All experiments were implemented in Pytorch 1.3.1 and Python 3.7.4 on Intel Xeon Gold 6230 processors. We used our own Pytorch implementation of CPO based on `https://github.com/jachiam/cpo`. For PPO, PPO Lagrangian, TRPO Lagrangian, we used an optimized PPO and TRPO implementation based on `https://github.com/Khrylx/PyTorch-RL`, `https://github.`

### G.1 Robots with Speed Limit

#### G.1.1 Environment Details

We used the MuJoCo environments provided by OpenAI Gym Brockman et al. (2016) for this set of experiments. For agents manuvering on a two-dimensional plane, the cost is calculated as

$$C(s, a) = \sqrt{v_x^2 + v_y^2}$$

For agents moving along a straight line, the cost is calculated as

$$C(s, a) = |v_x|$$

where $v_x, v_y$ are the velocities of the agent in the $x$ and $y$ directions respectively.

#### G.1.2 Algorithmic Hyperparameters

We used a two-layer feedforward neural network with a $\tanh$ activation for both our policy and value networks. We assume the policy is Gaussian with independent action dimensions. The policy networks outputs a mean vector and a vector containing the state-independent log standard deviations. States are normalized by the running mean the running standard deviation before being fed to any network. The advantage values are normalized by the batch mean and batch standard deviation before being used for policy updates. Except for the learning rate for $\nu$ which is kept fixed, all other learning rates are linearly annealed to 0 over the course of training. Our hyperparameter choices are based on the default choices in the implementations cited at the beginning of the section. For FOCOPS, PPO Lagrangian, and TRPO Lagrangian, we tuned the value of $\nu_{\max}$ across $\{1, 2, 3, 5, 10, +\infty\}$ and used the best value for each algorithm. However we found all three algorithms are not especially sensitive to the choice of $\nu_{\max}$. Table 3 summarizes the hyperparameters used in our experiments.

### G.2 Circle

#### G.2.1 Environment Details

In the circle tasks, the goal is for an agent to move along the circumference of a circle while remaining within a safety region smaller than the radius of the circle. The exact geometry of the task is shown in Figure 4. The reward and cost functions are defined as:

Figure 4: In the Circle task, reward is maximized by moving along the green circle. The agent is not allowed to enter the blue regions, so its optimal constrained path follows the line segments $AD$ and $BC$ (figure and caption taken from Achiam et al. (2017)).

$$R(s) = \frac{-yv_x + xv_y}{1 + |\sqrt{x^2 + y^2} - r|}$$
$$C(s) = \mathbf{1}(|x| > x_{\lim}).$$

Table 3: Hyperparameters for robots with speed limit experiments

| Hyperparameter | PPO | PPO-L | TRPO-L | CPO | FOCOPS |
|---|---|---|---|---|---|
| No. of hidden layers | 2 | 2 | 2 | 2 | 2 |
| No. of hidden nodes | 64 | 64 | 64 | 64 | 64 |
| Activation | tanh | tanh | tanh | tanh | tanh |
| Initial log std | -0.5 | -0.5 | -1 | -0.5 | -0.5 |
| Discount for reward $\gamma$ | 0.99 | 0.99 | 0.99 | 0.99 | 0.99 |
| Discount for cost $\gamma_C$ | 0.99 | 0.99 | 0.99 | 0.99 | 0.99 |
| Batch size | 2048 | 2048 | 2048 | 2048 | 2048 |
| Minibatch size | 64 | 64 | N/A | N/A | 64 |
| No. of optimization epochs | 10 | 10 | N/A | N/A | 10 |
| Maximum episode length | 1000 | 1000 | 1000 | 1000 | 1000 |
| GAE parameter (reward) | 0.95 | 0.95 | 0.95 | 0.95 | 0.95 |
| GAE parameter (cost) | N/A | 0.95 | 0.95 | 0.95 | 0.95 |
| Learning rate for policy | $3 \times 10^{-4}$ | $3 \times 10^{-4}$ | N/A | N/A | $3 \times 10^{-4}$ |
| Learning rate for reward value net | $3 \times 10^{-4}$ | $3 \times 10^{-4}$ | $3 \times 10^{-4}$ | $3 \times 10^{-4}$ | $3 \times 10^{-4}$ |
| Learning rate for cost value net | N/A | $3 \times 10^{-4}$ | $3 \times 10^{-4}$ | $3 \times 10^{-4}$ | $3 \times 10^{-4}$ |
| Learning rate for $\nu$ | N/A | 0.01 | 0.01 | N/A | 0.01 |
| $L2$-regularization coeff. for value net | $3 \times 10^{-3}$ | $3 \times 10^{-3}$ | $3 \times 10^{-3}$ | $3 \times 10^{-3}$ | $3 \times 10^{-3}$ |
| Clipping coefficient | 0.2 | 0.2 | N/A | N/A | N/A |
| Damping coeff. | N/A | N/A | 0.01 | 0.01 | N/A |
| Backtracking coeff. | N/A | N/A | 0.8 | 0.8 | N/A |
| Max backtracking iterations | N/A | N/A | 10 | 10 | N/A |
| Max conjugate gradient iterations | N/A | N/A | 10 | 10 | N/A |
| Iterations for training value net[1] | 1 | 1 | 80 | 80 | 1 |
| Temperature $\lambda$ | N/A | N/A | N/A | N/A | 1.5 |
| Trust region bound $\delta$ | N/A | N/A | 0.01 | 0.01 | 0.02 |
| Initial $\nu$, $\nu_{\max}$ | N/A | 0, 1 | 0, 2 | N/A | 0, 2 |

where $x, y$ are the positions of the agent on the plane, $v_x, v_y$ are the velocities of the agent along the $x$ and $y$ directions, $r$ is the radius of the circle, and $x_{\text{lim}}$ specifies the range of the safety region. The radius is set to $r = 10$ for both Ant and Humanoid while $x_{\text{lim}}$ is set to 3 and 2.5 for Ant and Humanoid respectively. Note that these settings are identical to those of the circle task in Achiam et al. (2017). Our experiments were implemented in OpenAI Gym (Brockman et al., 2016) while the circle tasks in Achiam et al. (2017) were implemented in rllab (Duan et al., 2016). We also excluded the Point agent from the original experiments since it is not a valid agent in OpenAI Gym. The first two dimensions in the state space are the $(x, y)$ coordinates of the center mass of the agent, hence the state space for both agents has two extra dimensions compared to the standard Ant and Humanoid environments from OpenAI Gym. Our open-source implementation of the circle environments can be found at `https://github.com/ymzhang01/mujoco-circle`.

### G.2.2 Algorithmic Hyperparameters

For these tasks, we used identical settings as the robots with speed limit tasks except we used a batch size of 50000 for all algorithms and a minibatch size of 1000 for PPO, PPO-Lagrangian, and FOCOPS. The discount rate for both reward and cost were set to 0.995. For FOCOPS, we set $\lambda = 1.0$ and $\delta = 0.04$.

## H   Generalization Analysis

We used trained agents using all four algorithms (PPO Lagrangian, TRPO Lagrangian, CPO, and FOCOPS) on robots with speed limit tasks shown in Figure 1. For each algorithm, we picked the seed with the highest maximum return of the last 100 episodes which does not violate the cost constraint at the end of training. The reasoning here is that for a fair comparison, we wish to pick the best performing seed for each algorithm. We then ran 10 episodes using the trained agents on 10

unseen random seeds (identical seeds are used for all four algorithms) to test how well the algorithms generalize over unseen data. The final results of running the trained agents on the speed limit and circle tasks are reported in Tables 4. We note that on unseen seeds FOCOPS outperforms the other three algorithms on five out of six tasks.

Table 4: Average return of 10 episodes for trained agents on the robots with speed limit tasks on 10 unseen random seeds. Results shown are the bootstrap mean and normal 95% confidence interval with 1000 bootstrap samples.

| Environment | | PPO-L | TRPO-L | CPO | FOCOPS |
|---|---|---|---|---|---|
| Ant-v3 | Reward | $920.4 \pm 75.9$ | $1721.4 \pm 191.2$ | $1335.57 \pm 43.17$ | $\mathbf{1934.9 \pm 99.5}$ |
| (103.12) | Cost | $68.25 \pm 11.05$ | $99.20 \pm 2.55$ | $80.72 \pm 3.82$ | $105.21 \pm 5.91$ |
| HalfCheetah-v3 | Reward | $1698.0 \pm 22.5$ | $1922.4 \pm 12.9$ | $1805.5 \pm 60.0$ | $\mathbf{2184.3 \pm 32.6}$ |
| (151.99) | Cost | $150.21 \pm 4.47$ | $179.82 \pm 1.73$ | $164.67 \pm 9.43$ | $158.39 \pm 6.56$ |
| Hopper-v3 | Reward | $2084.9 \pm 39.69$ | $2108.8 \pm 24.8$ | $\mathbf{2749.9 \pm 47.0}$ | $2446.2 \pm 9.0$ |
| (82.75) | Cost | $83.43 \pm 0.41$ | $82.17 \pm 1.53$ | $52.34 \pm 1.95$ | $81.26 \pm 0.88$ |
| Humanoid-v3 | Reward | $582.2 \pm 28.9$ | $3819.3 \pm 489.2$ | $1814.8 \pm 221.0$ | $\mathbf{4867.3 \pm 350.8}$ |
| (20.14) | Cost | $18.93 \pm 0.93$ | $18.60 \pm 1.27$ | $20.30 \pm 1.81$ | $21.58 \pm 0.74$ |
| Swimmer-v3 | Reward | $37.90 \pm 1.05$ | $33.48 \pm 0.44$ | $33.45 \pm 2.30$ | $\mathbf{39.37 \pm 2.04}$ |
| (24.52) | Cost | $25.49 \pm 0.57$ | $32.81 \pm 2.61$ | $22.61 \pm 0.33$ | $17.23 \pm 1.64$ |
| Walker2d-v3 | Reward | $1668.7 \pm 337.1$ | $2638.9 \pm 163.3$ | $2141.7 \pm 331.9$ | $\mathbf{3148.6 \pm 60.5}$ |
| (81.89) | Cost | $79.23 \pm 1.24$ | $90.96 \pm 0.97$ | $40.67 \pm 6.86$ | $73.35 \pm 2.67$ |

# I Sensitivity Analysis

We tested FOCOPS across ten different values of $\lambda$, and five difference values of $\nu_{\max}$ while keeping all other parameters fixed by running FOCOPS for 1 millon samples on each of the robots with speed limit experiment. For ease of comparison, we normalized the values by the return and cost of an unconstrained PPO agent trained for 1 million samples (i.e. if FOCOPS achieves a return of $x$ and an unconstrained PPO agent achieves a result of $y$, the normalized result reported is $x/y$) The results on the robots with speed limit tasks are reported in Tables 5 and 6. We note that the more challenging environments such as Humanoid are more sensitive to parameter choices but overall FOCOPS is largely insensitive to hyperparameter choices (especially the choice of $\nu_{\max}$). We also presented the performance of PPO-L and TRPO-L for different values of $\nu_{\max}$.

Table 5: Performance of FOCOPS for Different $\lambda$

| | Ant-v3 | | HalfCheetah-v3 | | Hopper-v3 | | Humanoid-v3 | | Swimmer-v3 | | Walker2d-v3 | | All Environments | |
|---|---|---|---|---|---|---|---|---|---|---|---|---|---|---|
| $\lambda$ | Reward | Cost | Reward | Cost | Reward | Cost | Reward | Cost | Reward | Cost | Reward | Cost | Reward | Cost |
| 0.1 | 0.66 | 0.55 | 0.38 | 0.46 | 0.77 | 0.50 | 0.63 | 0.52 | 0.34 | 0.51 | 0.43 | 0.48 | 0.53 | 0.50 |
| 0.5 | 0.77 | 0.54 | 0.38 | 0.45 | 0.97 | 0.50 | 0.71 | 0.54 | 0.36 | 0.50 | 0.66 | 0.50 | 0.64 | 0.50 |
| 1.0 | 0.83 | 0.55 | 0.47 | 0.47 | 1.04 | 0.50 | 0.80 | 0.52 | 0.34 | 0.49 | 0.76 | 0.49 | 0.70 | 0.50 |
| 1.3 | 0.83 | 0.55 | 0.42 | 0.47 | 1.00 | 0.50 | 0.85 | 0.53 | 0.36 | 0.51 | 0.87 | 0.49 | 0.72 | 0.51 |
| 1.5 | 0.83 | 0.55 | 0.42 | 0.47 | 1.01 | 0.50 | 0.87 | 0.52 | 0.37 | 0.51 | 0.87 | 0.50 | 0.73 | 0.51 |
| 2.0 | 0.83 | 0.55 | 0.42 | 0.47 | 1.06 | 0.50 | 0.89 | 0.52 | 0.37 | 0.52 | 0.82 | 0.45 | 0.73 | 0.51 |
| 2.5 | 0.79 | 0.54 | 0.43 | 0.47 | 1.03 | 0.50 | 0.94 | 0.53 | 0.35 | 0.50 | 0.73 | 0.49 | 0.71 | 0.51 |
| 3.0 | 0.76 | 0.54 | 0.42 | 0.47 | 1.01 | 0.49 | 0.92 | 0.52 | 0.41 | 0.50 | 0.77 | 0.49 | 0.72 | 0.50 |
| 4.0 | 0.70 | 0.54 | 0.40 | 0.46 | 1.00 | 0.49 | 0.87 | 0.53 | 0.43 | 0.49 | 0.64 | 0.49 | 0.67 | 0.50 |
| 5.0 | 0.64 | 0.55 | 0.40 | 0.47 | 1.01 | 0.50 | 0.81 | 0.54 | 0.38 | 0.49 | 0.57 | 0.50 | 0.63 | 0.51 |

Table 6: Performance of FOCOPS for Different $\nu_{\max}$

| $\nu_{\max}$ | Ant-v3 | | HalfCheetah-v3 | | Hopper-v3 | | Humanoid-v3 | | Swimmer-v3 | | Walker2d-v3 | | All Environments | |
|---|---|---|---|---|---|---|---|---|---|---|---|---|---|---|
| | Reward | Cost | Reward | Cost | Reward | Cost | Reward | Cost | Reward | Cost | Reward | Cost | Reward | Cost |
| 1 | 0.83 | 0.55 | 0.45 | 0.61 | 1.00 | 0.51 | 0.87 | 0.52 | 0.40 | 0.62 | 0.88 | 0.50 | 0.74 | 0.55 |
| 2 | 0.83 | 0.55 | 0.42 | 0.47 | 1.01 | 0.50 | 0.87 | 0.52 | 0.35 | 0.51 | 0.87 | 0.50 | 0.73 | 0.51 |
| 3 | 0.81 | 0.54 | 0.41 | 0.47 | 1.01 | 0.49 | 0.83 | 0.53 | 0.34 | 0.49 | 0.87 | 0.50 | 0.71 | 0.50 |
| 5 | 0.82 | 0.55 | 0.41 | 0.47 | 1.01 | 0.50 | 0.83 | 0.53 | 0.31 | 0.49 | 0.87 | 0.50 | 0.71 | 0.51 |
| 10 | 0.82 | 0.55 | 0.41 | 0.47 | 1.01 | 0.50 | 0.83 | 0.53 | 0.34 | 0.47 | 0.87 | 0.50 | 0.71 | 0.50 |
| $+\infty$ | 0.82 | 0.55 | 0.41 | 0.47 | 1.01 | 0.50 | 0.83 | 0.53 | 0.35 | 0.47 | 0.88 | 0.50 | 0.72 | 0.50 |

Table 7: Performance of PPO Lagrangian for Different $\nu_{\max}$

| $\nu_{\max}$ | Ant-v3 | | HalfCheetah-v3 | | Hopper-v3 | | Humanoid-v3 | | Swimmer-v3 | | Walker2d-v3 | | All Environments | |
|---|---|---|---|---|---|---|---|---|---|---|---|---|---|---|
| | Reward | Cost | Reward | Cost | Reward | Cost | Reward | Cost | Reward | Cost | Reward | Cost | Reward | Cost |
| 1 | 0.80 | 0.55 | 0.41 | 0.49 | 0.98 | 0.49 | 0.73 | 0.52 | 0.28 | 0.50 | 0.77 | 0.50 | 0.66 | 0.51 |
| 2 | 0.71 | 0.49 | 0.36 | 0.50 | 0.81 | 0.48 | 0.73 | 0.52 | 0.32 | 0.50 | 0.72 | 0.50 | 0.61 | 0.50 |
| 3 | 0.78 | 0.54 | 0.36 | 0.47 | 0.73 | 0.49 | 0.73 | 0.52 | 0.40 | 0.48 | 0.72 | 0.50 | 0.62 | 0.50 |
| 5 | 0.77 | 0.53 | 0.35 | 0.47 | 0.73 | 0.49 | 0.73 | 0.52 | 0.40 | 0.49 | 0.72 | 0.50 | 0.62 | 0.50 |
| 10 | 0.77 | 0.54 | 0.36 | 0.47 | 0.73 | 0.49 | 0.73 | 0.52 | 0.40 | 0.49 | 0.72 | 0.50 | 0.62 | 0.50 |
| $+\infty$ | 0.66 | 0.54 | 0.27 | 0.45 | 0.73 | 0.49 | 0.55 | 0.47 | 0.40 | 0.49 | 0.72 | 0.50 | 0.55 | 0.49 |

Table 8: Performance of TRPO Lagrangian for Different $\nu_{\max}$

| $\nu_{\max}$ | Ant-v3 | | HalfCheetah-v3 | | Hopper-v3 | | Humanoid-v3 | | Swimmer-v3 | | Walker2d-v3 | | All Environments | |
|---|---|---|---|---|---|---|---|---|---|---|---|---|---|---|
| | Reward | Cost | Reward | Cost | Reward | Cost | Reward | Cost | Reward | Cost | Reward | Cost | Reward | Cost |
| 1 | 0.71 | 0.50 | 0.70 | 0.68 | 0.61 | 0.50 | 0.68 | 0.50 | 0.43 | 0.61 | 0.48 | 0.50 | 0.61 | 0.55 |
| 2 | 0.70 | 0.51 | 0.50 | 0.53 | 0.39 | 0.53 | 0.68 | 0.50 | 0.33 | 0.53 | 0.36 | 0.50 | 0.49 | 0.52 |
| 3 | 0.70 | 0.51 | 0.52 | 0.53 | 0.41 | 0.53 | 0.68 | 0.50 | 0.30 | 0.67 | 0.35 | 0.50 | 0.49 | 0.54 |
| 5 | 0.70 | 0.51 | 0.49 | 0.52 | 0.36 | 0.52 | 0.68 | 0.50 | 0.23 | 0.67 | 0.35 | 0.51 | 0.47 | 0.54 |
| 10 | 0.70 | 0.51 | 0.48 | 0.51 | 0.34 | 0.52 | 0.68 | 0.50 | 0.31 | 0.77 | 0.34 | 0.50 | 0.47 | 0.55 |
| $+\infty$ | 0.70 | 0.51 | 0.48 | 0.51 | 0.36 | 0.52 | 0.68 | 0.50 | 0.30 | 0.78 | 0.34 | 0.50 | 0.48 | 0.55 |

[Supplementary Material 2]

# Supplementary Material for First Order Constrained Optimization in Policy Space

# Appendices

## A Proof of Theorem 1

**Theorem 1.** *Let $\tilde{b} = (1-\gamma)(b - \tilde{J}_C(\pi_{\theta_k}))$. If $\pi_{\theta_k}$ is a feasible solution for (4-6) takes the form*

$$\pi^*(a|s) = \frac{\pi_{\theta_k}(a|s)}{Z_{\lambda,\nu}(s)} \exp\left(\frac{1}{\lambda}\left(A^{\pi_{\theta_k}}(s,a) - \nu A_C^{\pi_{\theta_k}}(s,a)\right)\right) \qquad (7)$$

*where $Z_{\lambda,\nu}(s)$ is the partition function which ensures (7) is a valid probability distribution, $\lambda$ and $\nu$ are solutions to the optimization problem:*

$$\min_{\lambda,\nu \geq 0} \lambda\delta + \nu\tilde{b} + \lambda \mathop{\mathbb{E}}_{\substack{s \sim d^{\pi_{\theta_k}} \\ a \sim \pi^*}} \left[\log Z_{\lambda,\nu}(s)\right] \qquad (8)$$

*Proof.* We will begin by showing that Problem (4-6) is convex w.r.t. $\pi = \{\pi(a|s) : s \in \mathcal{S}, a \in \mathcal{A}\}$. First note that the objective function is linear w.r.t. $\pi$. Since $J_C(\pi_{\theta_k})$ is a constant w.r.t. $\pi$, constraint (5) is linear. Constraint (6) can be rewritten as $\sum_s d^{\pi_{\theta_k}}(s) D_{\text{KL}}\left(\pi\|\pi_{\theta_k}\right)[s] \leq \delta$, the KL divergence is convex w.r.t. its first argument, therefore constraint (6) which is a linear combination of convex functions is also convex. Since $\pi_{\theta_k}$ satisfies Constraint (5) and is also an interior point within the set given by Constraints (6) ($D_{\text{KL}}\left(\pi_{\theta_k}\|\pi_{\theta_k}\right) = 0$, and $\delta > 0$), therefore Slater's constraint qualification holds, strong duality holds.

We can therefore solve for the optimal value of Problem (4-6) $p^*$ by solving the corresponding dual problem. Let

$$L(\pi,\lambda,\nu) = \lambda\delta + \nu\tilde{b} + \mathop{\mathbb{E}}_{s \sim d^{\pi_{\theta_k}}}\left[\mathop{\mathbb{E}}_{a \sim \pi(\cdot|s)}[A^{\pi_{\theta_k}}(s,a)] - \nu\mathop{\mathbb{E}}_{a \sim \pi(\cdot|s)}[A_C^{\pi_{\theta_k}}(s,a)] - \lambda D_{\text{KL}}\left(\pi\|\pi_{\theta_k}\right)[s]\right] \quad (16)$$

Therefore,

$$p^* = \max_{\pi \in \Pi} \min_{\lambda,\nu \geq 0} L(\pi,\lambda,\nu) = \min_{\lambda,\nu \geq 0} \max_{\pi \in \Pi} L(\pi,\lambda,\nu) \qquad (17)$$

where we invoked strong duality in the second equality. We note that if $\pi^*, \lambda^*, \nu^*$ are optimal for (17), $\pi^*$ is also optimal for Problem (4-6) (Boyd and Vandenberghe, 2004).

Consider the inner maximization problem in (17), we can decompose this problem into separate problems, one for each $s$. This gives us an optimization problem of the form,

$$
\begin{aligned}
\underset{\pi}{\text{maximize}} \quad & \mathop{\mathbb{E}}_{a \sim \pi(\cdot|s)}\left[A^{\pi_{\theta_k}}(s,a) - \nu A_C^{\pi_{\theta_k}}(s,a) - \lambda(\log\pi(a|s) - \log\pi_{\theta_k}(a|s))\right] \\
\text{subject to} \quad & \sum_a \pi(a|s) = 1 \\
& \pi(a|s) \geq 0 \quad \text{for all } a \in \mathcal{A}
\end{aligned}
\qquad (18)
$$

which is equivalent to the inner maximization problem in (17). This is clearly a convex optimization problem which we can solve using a simple Lagrangian argument. We can write the Lagrangian of (18) as

$$G(\pi) = \sum_a \pi(a|s)\left[A^{\pi_{\theta_k}}(s,a) - \nu A_C^{\pi_{\theta_k}}(s,a) - \lambda(\log\pi(a|s) - \log\pi_{\theta_k}(a|s)) + \zeta\right] - 1 \quad (19)$$

where $\zeta > 0$ is the Lagrange multiplier associated with the constraint $\sum_a \pi(a|s) = 1$. Differentiating $G(\pi)$ w.r.t. $\pi(a|s)$ for some $a$:

$$\frac{\partial G}{\partial \pi(a|s)} = A^{\pi_{\theta_k}}(s,a) - \nu A_C^{\pi_{\theta_k}}(s,a) - \lambda(\log \pi(a|s) + 1 - \log \pi_{\theta_k}(a|s)) + \zeta \qquad (20)$$

Setting (20) to zero and rearranging the term, we obtain

$$\pi(a|s) = \pi_{\theta_k}(a|s) \exp\left(\frac{1}{\lambda}\left(A^{\pi_{\theta_k}}(s,a) - \nu A_C^{\pi_{\theta_k}}(s,a)\right) + \frac{\zeta}{\lambda} + 1\right) \qquad (21)$$

We chose $\zeta$ so that $\sum_a \pi(a|s) = 1$ and rewrite $\zeta/\lambda + 1$ as $Z_{\lambda,\nu}(s)$. We find that the optimal solution $\pi^*$ to (18) takes the form

$$\pi^*(a|s) = \frac{\pi_{\theta_k}(a|s)}{Z_{\lambda,\nu}(s)} \exp\left(\frac{1}{\lambda}\left(A^{\pi_{\theta_k}}(s,a) - \nu A_C^{\pi_{\theta_k}}(s,a)\right)\right)$$

Plugging $\pi^*$ back into Equation 17 gives us

$$
\begin{aligned}
p^* &= \min_{\lambda,\nu \geq 0} \lambda\delta + \nu\tilde{b} + \mathop{\mathbb{E}}_{\substack{s \sim d^{\pi_{\theta_k}} \\ a \sim \pi^*}} [A^{\pi_{\theta_k}}(s,a) - \nu A_C^{\pi_{\theta_k}}(s,a) - \lambda(\log \pi^*(a|s) - \log \pi_{\theta_k}(a|s))] \\
&= \min_{\lambda,\nu \geq 0} \lambda\delta + \nu\tilde{b} + \mathop{\mathbb{E}}_{\substack{s \sim d^{\pi_{\theta_k}} \\ a \sim \pi^*}} [A^{\pi_{\theta_k}}(s,a) - \nu A_C^{\pi_{\theta_k}}(s,a) - \lambda(\log \pi_{\theta_k}(a|s) - \log Z_{\lambda,\nu}(s) \\
&\quad + \frac{1}{\lambda}(A^{\pi_{\theta_k}}(s,a) - \nu A_C^{\pi_{\theta_k}}(s,a)) - \log \pi_{\theta_k}(a|s))] \\
&= \min_{\lambda,\nu \geq 0} \lambda\delta + \nu\tilde{b} + \lambda \mathop{\mathbb{E}}_{\substack{s \sim d^{\pi_{\theta_k}} \\ a \sim \pi^*}} [\log Z_{\lambda,\nu}(s)]
\end{aligned}
$$

$\square$

# B  Proof of Corollary 1

**Corollary 1.** *The gradient of $\mathcal{L}(\theta)$ takes the form*

$$\nabla_\theta \mathcal{L}(\theta) = \mathop{\mathbb{E}}_{s \sim d^{\pi_{\theta_k}}} \left[\nabla_\theta D_{KL}\left(\pi_\theta \| \pi^*\right)[s]\right] \qquad (11)$$

*where*

$$\nabla_\theta D_{KL}\left(\pi_\theta \| \pi^*\right)[s] = \nabla_\theta D_{KL}\left(\pi_\theta \| \pi_{\theta_k}\right)[s] - \frac{1}{\lambda}\mathop{\mathbb{E}}_{a \sim \pi_{\theta_k}}\left[\frac{\nabla_\theta \pi_\theta(a|s)}{\pi_{\theta_k}(a|s)}\left(A^{\pi_{\theta_k}}(s,a) - \nu A_C^{\pi_{\theta_k}}(s,a)\right)\right] \quad (12)$$

*Proof.* We only need to calculate the gradient of the loss function for a single sampled $s$. We first note that,

$$D_{\text{KL}}\left(\pi_\theta \| \pi^*\right)[s] = -\sum_a \pi_\theta(a|s) \log \pi^*(a|s) + \sum_a \pi_\theta(a|s) \log \pi_\theta(a|s)$$

$$= H(\pi_\theta, \pi^*)[s] - H(\pi_\theta)[s]$$

where $H(\pi_\theta)[s]$ is the entropy and $H(\pi_\theta, \pi^*)[s]$ is the cross-entropy under state $s$. We expand the cross entropy term which gives us

$$
\begin{aligned}
H(\pi_\theta, \pi^*)[s] &= -\sum_a \pi_\theta(a|s) \log \pi^*(a|s) \\
&= -\sum_a \pi_\theta(a|s) \log\left(\frac{\pi_{\theta_k}(a|s)}{Z_{\lambda,\nu}(s)} \exp\left[\frac{1}{\lambda}\left(A^{\pi_{\theta_k}}(s,a) - \nu A_C^{\pi_{\theta_k}}(s,a)\right)\right]\right) \\
&= -\sum_a \pi_\theta(a|s) \log \pi_{\theta_k}(a|s) + \log Z_{\lambda,\nu}(s) - \frac{1}{\lambda}\sum_a \pi_\theta(a|s)\left(A^{\pi_{\theta_k}}(s,a) - \nu A_C^{\pi_{\theta_k}}(s,a)\right)
\end{aligned}
$$

We then subtract the entropy term to recover the KL divergence:

$$D_{\mathrm{KL}}\left(\pi_\theta \| \pi^*\right)[s] = D_{\mathrm{KL}}\left(\pi_\theta \| \pi_{\theta_k}\right)[s] + \log Z_{\lambda,\nu}(s) - \frac{1}{\lambda}\sum_a \pi_\theta(a|s)\left(A^{\pi_{\theta_k}}(s,a) - \nu A_C^{\pi_{\theta_k}}(s,a)\right)$$

$$= D_{\mathrm{KL}}\left(\pi_\theta \| \pi_{\theta_k}\right)[s] + \log Z_{\lambda,\nu}(s) - \frac{1}{\lambda}\mathop{\mathbb{E}}_{a\sim\pi_{\theta_k}(\cdot|s)}\left[\frac{\pi_\theta(a|s)}{\pi_{\theta_k}(a|s)}\left(A^{\pi_{\theta_k}}(s,a) - \nu A_C^{\pi_{\theta_k}}(s,a)\right)\right]$$

where in the last equality we applied importance sampling to rewrite the expectation w.r.t. $\pi_{\theta_k}$. Finally, taking the gradient on both sides gives us:

$$\nabla_\theta D_{\mathrm{KL}}\left(\pi_\theta \| \pi^*\right)[s] = \nabla_\theta D_{\mathrm{KL}}\left(\pi_\theta \| \pi_{\theta_k}\right)[s] - \frac{1}{\lambda}\mathop{\mathbb{E}}_{a\sim\pi_{\theta_k}(\cdot|s)}\left[\frac{\nabla_\theta\pi_\theta(a|s)}{\pi_{\theta_k}(a|s)}\left(A^{\pi_{\theta_k}}(s,a) - \nu A_C^{\pi_{\theta_k}}(s,a)\right)\right].$$

$\square$

## C   Proof of Corollary 2

**Corollary 2.** *The derivative of $L(\pi^*, \lambda, \nu)$ w.r.t. $\nu$ is*

$$\frac{\partial L(\pi^*, \lambda, \nu)}{\partial \nu} = \tilde{b} - \mathop{\mathbb{E}}_{\substack{s\sim d^{\pi_{\theta_k}}\\a\sim\pi^*}}\left[A^{\pi_{\theta_k}}(s,a)\right] \tag{13}$$

*Proof.* From Theorem 1, we have

$$L(\pi^*, \lambda, \nu) = \lambda\delta + \nu\tilde{b} + \lambda \mathop{\mathbb{E}}_{\substack{s\sim d^{\pi_{\theta_k}}\\a\sim\pi^*}}\left[\log Z_{\lambda,\nu}(s)\right]. \tag{22}$$

The first two terms is an affine function w.r.t. $\nu$, therefore its derivative is $\tilde{b}$. We will then focus on the expectation in the last term. To simplify our derivation, we will first calculate the derivative of $\pi^*$ w.r.t. $\nu$,

$$\frac{\partial \pi^*(a|s)}{\partial \nu} = \frac{\pi_{\theta_k}(a|s)}{Z_{\lambda,\nu}^2(s)}\left[Z_{\lambda,\nu}(s)\frac{\partial}{\partial\nu}\exp\left(\frac{1}{\lambda}\left(A^{\pi_{\theta_k}}(s,a) - \nu A_C^{\pi_{\theta_k}}(s,a)\right)\right)\right.$$

$$\left. - \exp\left(\frac{1}{\lambda}\left(A^{\pi_{\theta_k}}(s,a) - \nu A_C^{\pi_{\theta_k}}(s,a)\right)\right)\frac{\partial Z_{\lambda,\nu}(s)}{\partial\nu}\right]$$

$$= -\frac{A_C^{\pi_{\theta_k}}(s,a)}{\lambda}\pi^*(a|s) - \pi^*(a|s)\frac{\partial \log Z_{\lambda,\nu}(s)}{\partial\nu}$$

Therefore the derivative of the expectation in the last term of $L(\pi^*, \lambda, \nu)$ can be written as

$$\frac{\partial}{\partial\nu}\mathop{\mathbb{E}}_{\substack{s\sim d^{\pi_{\theta_k}}\\a\sim\pi^*}}\left[\log Z_{\lambda,\nu}(s)\right]$$

$$= \mathop{\mathbb{E}}_{\substack{s\sim d^{\pi_{\theta_k}}\\a\sim\pi_{\theta_k}}}\left[\frac{\partial}{\partial\nu}\left(\frac{\pi^*(a|s)}{\pi_{\theta_k}(a|s)}\log Z_{\lambda,\nu}(s)\right)\right]$$

$$= \mathop{\mathbb{E}}_{\substack{s\sim d^{\pi_{\theta_k}}\\a\sim\pi_{\theta_k}}}\left[\frac{1}{\pi_{\theta_k}(a|s)}\left(\frac{\partial\pi^*(a|s)}{\partial\nu}\log Z_{\lambda,\nu}(s) + \pi^*(a|s)\frac{\partial \log Z_{\lambda,\nu}(s)}{\partial\nu}\right)\right]$$

$$= \mathop{\mathbb{E}}_{\substack{s\sim d^{\pi_{\theta_k}}\\a\sim\pi_{\theta_k}}}\left[\frac{\pi^*(a|s)}{\pi_{\theta_k}(a|s)}\left(-\frac{A_C^{\pi_{\theta_k}}(s,a)}{\lambda}\log Z_{\lambda,\nu}(s) - \frac{\partial \log Z_{\lambda,\nu}(s)}{\partial\nu}\log Z_{\lambda,\nu}(s) + \frac{\partial \log Z_{\lambda,\nu}(s)}{\partial\nu}\right)\right]$$

$$= \mathop{\mathbb{E}}_{\substack{s\sim d^{\pi_{\theta_k}}\\a\sim\pi^*}}\left[-\frac{A_C^{\pi_{\theta_k}}(s,a)}{\lambda}\log Z_{\lambda,\nu}(s) - \frac{\partial \log Z_{\lambda,\nu}(s)}{\partial\nu}\log Z_{\lambda,\nu}(s) + \frac{\partial \log Z_{\lambda,\nu}(s)}{\partial\nu}\right].$$

$$\tag{23}$$

Also,

$$
\begin{aligned}
\frac{\partial Z_{\lambda,\nu}(s)}{\partial \nu} &= \frac{\partial}{\partial \nu} \sum_a \pi_{\theta_k}(a|s) \exp\left(\frac{1}{\lambda}\left(A^{\pi_{\theta_k}}(s,a) - \nu A_C^{\pi_{\theta_k}}(s,a)\right)\right) \\
&= \sum_a -\pi_{\theta_k}(a|s) \frac{A_C^{\pi_{\theta_k}}(s,a)}{\lambda} \exp\left(\frac{1}{\lambda}\left(A^{\pi_{\theta_k}}(s,a) - \nu A_C^{\pi_{\theta_k}}(s,a)\right)\right) \\
&= \sum_a -\frac{A_C^{\pi_{\theta_k}}(s,a)}{\lambda} \frac{\pi_{\theta_k}(a|s)}{Z_{\lambda,\nu}(s)} \exp\left(\frac{1}{\lambda}\left(A^{\pi_{\theta_k}}(s,a) - \nu A_C^{\pi_{\theta_k}}(s,a)\right)\right) Z_{\lambda,\nu}(s) \\
&= -\frac{Z_{\lambda,\nu}(s)}{\lambda} \mathop{\mathbb{E}}_{a\sim\pi^*(\cdot|s)}\left[A_C^{\pi_{\theta_k}}(s,a)\right].
\end{aligned}
\tag{24}
$$

Therefore,

$$
\frac{\partial \log Z_{\lambda,\nu}(s)}{\partial \nu} = \frac{\partial Z_{\lambda,\nu}(s)}{\partial \nu} \frac{1}{Z_{\lambda,\nu}(s)} = -\frac{1}{\lambda} \mathop{\mathbb{E}}_{a\sim\pi^*(\cdot|s)}\left[A_C^{\pi_{\theta_k}}(s,a)\right].
\tag{25}
$$

Plugging (25) into the last equality in (23) gives us

$$
\begin{aligned}
\frac{\partial}{\partial \nu} \mathop{\mathbb{E}}_{\substack{s\sim d^{\pi_{\theta_k}} \\ a\sim\pi^*}}[\log Z_{\lambda,\nu}(s)] &= \mathop{\mathbb{E}}_{\substack{s\sim d^{\pi_{\theta_k}} \\ a\sim\pi^*}}\left[-\frac{A_C^{\pi_{\theta_k}}(s,a)}{\lambda}\log Z_{\lambda,\nu}(s) + \frac{A_C^{\pi_{\theta_k}}(s,a)}{\lambda}\log Z_{\lambda,\nu}(s) - \frac{1}{\lambda}A_C^{\pi_{\theta_k}}(s,a)\right] \\
&= -\frac{1}{\lambda}\mathop{\mathbb{E}}_{\substack{s\sim d^{\pi_{\theta_k}} \\ a\sim\pi^*}}\left[A_C^{\pi_{\theta_k}}(s,a)\right].
\end{aligned}
\tag{26}
$$

Combining (26) with the derivatives of the affine term gives us the final desired result.

$\square$

# D  PPO Lagrangian and TRPO Lagrangian

## D.1  PPO-Lagrangian

Recall that the PPO (clipped) objective takes the form (Schulman et al., 2017b)

$$
L(\theta) = \min\left(\frac{\pi_\theta(a|s)}{\pi_{\theta_k}(a|s)}A^{\pi_{\theta_k}}(s,a), \text{clip}\left(\frac{\pi_\theta(a|s)}{\pi_{\theta_k}(a|s)}, 1-\epsilon, 1+\epsilon\right)A^{\pi_{\theta_k}}(s,a)\right)
\tag{27}
$$

We augment this objective with an additional term to form the a new objective function

$$
\tilde{L}(\theta) = L(\theta) + \nu\left(J_C(\pi_\theta) - b\right)
\tag{28}
$$

The Lagrangian method involves a maximization and a minimization step. For the maximization step, we optimize the objective (28) by performing backpropagation w.r.t. $\theta$. For the minimization step, we apply gradient descent to the same objective w.r.t $\nu$. Like FOCOPS, the PPO-Lagrangian algorithm is also first-order thus simple to implement, however its empirical performance deteriorates drastically on more challenging environments (See Section 4). Furthermore, it remains an open question whether PPO-Lagrangian satisfies any worst-case constraint guarantees.

## D.2  TRPO-Lagrangian

Similar to the PPO-Lagrangian method, we instead optimize an augmented TRPO problem

$$
\underset{\theta}{\text{maximize}} \quad \tilde{L}(\theta)
\tag{29}
$$

$$
\text{subject to} \quad \bar{D}_{\text{KL}}(\pi_\theta \,\|\, \pi_{\theta_k})[s] \leq \delta.
\tag{30}
$$

where

$$
\tilde{L}(\theta) = \frac{\pi_\theta(a|s)}{\pi_{\theta_k}(a|s)}A^{\pi_{\theta_k}}(s,a) + \nu\left(J_C(\pi_\theta) - b\right)
\tag{31}
$$

We then apply Taylor approximation to (29) and (30) which gives us

$$\underset{\theta}{\text{maximize}} \quad \tilde{g}^T(\theta - \theta_k) \tag{32}$$

$$\text{subject to} \quad \frac{1}{2}(\theta - \theta_k)^T H(\theta - \theta_k) \leq \delta. \tag{33}$$

Here $\tilde{g}$ is the gradient of (29) w.r.t. the parameter $\theta$.

Like for the PPO-Lagrangian method, we first perform a maximization step where we optimize (32)-(33) using the TRPO method (Schulman et al., 2015). We then perform a minimization step by updating $\nu$. TRPO-Lagrangian is also a second-order algorithm. Performance-wise TRPO-Lagrangian does poorly in terms of constraint satisfaction. Like PPO-Lagrangian, further research into the theoretical properties of TRPO merits further research.

## E    FOCOPS for Different Cost Thresholds

In this section, we verify that FOCOPS works effectively for different threshold levels. We experiment on the robots with speed limits environments. For each environment, we calculated the cost required for an unconstrained PPO agent after training for 1 million samples. We then used 25%, 50%, and 75% of this cost as our cost thresholds and trained FOCOPS on each of thresholds respectively. The learning curves are reported in Figure 3. We note from these plots that FOCOPS can effectively learn constraint-satisfying policies for different cost thresholds.

Figure 3: Performance of FOCOPS on robots with speed limit tasks with different cost thresholds. The $x$-axis represent the number of samples used and the $y$-axis represent the average total reward/cost return of the last 100 episodes. The solid line represent the mean of 1000 bootstrap samples over 10 random seeds. The horizontal lines in the cost plots represent the cost thresholds corresponding to 25%, 50%, and 75% of the cost required by an unconstrained PPO agent trained with 1 million samples. Each solid line represents FOCOPS trained with the corresponding thresholds. The shaded regions represent the bootstrap normal 95% confidence interval. Each of the solid lines represent

## F Pseudocode

---

**Algorithm 2** First Order Constrained Optimization in Policy Space (FOCOPS)

---

**Initialize:** Policy network $\pi_\theta$; Value network for return $V_\phi$; Value network for costs $V_\psi^C$.

**Initialize:** Discount rates $\gamma$, GAE parameter $\beta$; Learning rates $\alpha_\nu, \alpha_V, \alpha_\pi$; Temperature $\lambda$; Initial cost constraint parameter $\nu$; Cost constraint parameter bound $\nu_{\max}$. Trust region bound $\delta$; Cost bound $b$.

  **while** Stopping criteria not met **do**

    Generate batch data of $M$ episodes of length $T$ from $(s_{i,t}, a_{i,t}, r_{i,t}, s_{i,t+1}, c_{i,t})$ from $\pi_\theta$, $i = 1, \ldots, M, t = 1, \ldots, T$.

    Estimate $C$-return by averaging over $C$-return for all episodes:

$$\hat{J}_C = \frac{1}{M} \sum_{i=1}^{M} \sum_{t=0}^{T-1} \gamma^t c_{i,t}$$

    Store old policy $\theta' \leftarrow \theta$

    Estimate advantage functions $\hat{A}_{i,t}$ and $\hat{A}_{i,t}^C$, $i = 1, \ldots, M, t = 1, \ldots, T$ using GAE.

    Get $V_{i,t}^{\text{target}} = \hat{A}_{i,t} + V_\phi(s_{i,t})$ and $V_{i,t}^{C,\text{target}} = \hat{A}_{i,t} + V_\psi^C(s_{i,t})$

    Update $\nu$ by

$$\nu \leftarrow \operatorname*{proj}_\nu \left[ \nu - \alpha_\nu \left( b - \hat{J}_C \right) \right]$$

    **for** $K$ epochs **do**

      **for** each minibatch $\{s_j, a_j, A_j, A_j^C, V_j^{\text{target}}, V_j^{C,\text{target}}\}$ of size $B$ **do**

        Value loss functions

$$\mathcal{L}_V(\phi) = \frac{1}{2N} \sum_{j=1}^{B} (V_\phi(s_j) - V_j^{\text{target}})^2$$

$$\mathcal{L}_{V^C}(\psi) = \frac{1}{2N} \sum_{j=1}^{B} (V_\psi(s_j) - V_j^{C,\text{target}})^2$$

        Update value networks

$$\phi \leftarrow \phi - \alpha_V \nabla_\phi \mathcal{L}_V(\phi)$$
$$\psi \leftarrow \psi - \alpha_V \nabla_\psi \mathcal{L}_{V^C}(\psi)$$

        Update policy

$$\theta \leftarrow \theta - \alpha_\pi \hat{\nabla}_\theta \mathcal{L}_\pi(\theta)$$

        where

$$\hat{\nabla}_\theta \mathcal{L}_\pi(\theta) \approx \frac{1}{B} \sum_{j=1}^{B} \left[ \nabla_\theta D_{\text{KL}} \left( \pi_\theta \| \pi_{\theta'} \right) [s_j] - \frac{1}{\lambda} \frac{\nabla_\theta \pi_\theta(a_j|s_j)}{\pi_{\theta'}(a_j|s_j)} \left( \hat{A}_j - \nu \hat{A}_j^C \right) \right] \mathbf{1}_{D_{\text{KL}}(\pi_\theta \| \pi_{\theta'})[s_j] \leq \delta}$$

      **if** $\frac{1}{MT} \sum_{i=1}^{M} \sum_{t=0}^{T-1} D_{\text{KL}} \left( \pi_\theta \| \pi_{\theta'} \right) [s_{i,t}] > \delta$ **then**

        Break out of inner loop

---

## G Implementation Details for Experiments

Our open-source implementation of FOCOPS can be found at `https://github.com/ymzhang01/focops`. All experiments were implemented in Pytorch 1.3.1 and Python 3.7.4 on Intel Xeon Gold 6230 processors. We used our own Pytorch implementation of CPO based on `https://github.com/jachiam/cpo`. For PPO, PPO Lagrangian, TRPO Lagrangian, we used an optimized PPO and TRPO implementation based on `https://github.com/Khrylx/PyTorch-RL`, `https://github.`

### G.1 Robots with Speed Limit

#### G.1.1 Environment Details

We used the MuJoCo environments provided by OpenAI Gym Brockman et al. (2016) for this set of experiments. For agents manuvering on a two-dimensional plane, the cost is calculated as

$$C(s, a) = \sqrt{v_x^2 + v_y^2}$$

For agents moving along a straight line, the cost is calculated as

$$C(s, a) = |v_x|$$

where $v_x, v_y$ are the velocities of the agent in the $x$ and $y$ directions respectively.

#### G.1.2 Algorithmic Hyperparameters

We used a two-layer feedforward neural network with a $\tanh$ activation for both our policy and value networks. We assume the policy is Gaussian with independent action dimensions. The policy networks outputs a mean vector and a vector containing the state-independent log standard deviations. States are normalized by the running mean the running standard deviation before being fed to any network. The advantage values are normalized by the batch mean and batch standard deviation before being used for policy updates. Except for the learning rate for $\nu$ which is kept fixed, all other learning rates are linearly annealed to 0 over the course of training. Our hyperparameter choices are based on the default choices in the implementations cited at the beginning of the section. For FOCOPS, PPO Lagrangian, and TRPO Lagrangian, we tuned the value of $\nu_{\max}$ across $\{1, 2, 3, 5, 10, +\infty\}$ and used the best value for each algorithm. However we found all three algorithms are not especially sensitive to the choice of $\nu_{\max}$. Table 3 summarizes the hyperparameters used in our experiments.

### G.2 Circle

#### G.2.1 Environment Details

In the circle tasks, the goal is for an agent to move along the circumference of a circle while remaining within a safety region smaller than the radius of the circle. The exact geometry of the task is shown in Figure 4. The reward and cost functions are defined as:

Figure 4: In the Circle task, reward is maximized by moving along the green circle. The agent is not allowed to enter the blue regions, so its optimal constrained path follows the line segments $AD$ and $BC$ (figure and caption taken from Achiam et al. (2017)).

$$R(s) = \frac{-yv_x + xv_y}{1 + |\sqrt{x^2 + y^2} - r|}$$

$$C(s) = \mathbf{1}(|x| > x_{\lim}).$$

Table 3: Hyperparameters for robots with speed limit experiments

| Hyperparameter | PPO | PPO-L | TRPO-L | CPO | FOCOPS |
|---|---|---|---|---|---|
| No. of hidden layers | 2 | 2 | 2 | 2 | 2 |
| No. of hidden nodes | 64 | 64 | 64 | 64 | 64 |
| Activation | tanh | tanh | tanh | tanh | tanh |
| Initial log std | -0.5 | -0.5 | -1 | -0.5 | -0.5 |
| Discount for reward $\gamma$ | 0.99 | 0.99 | 0.99 | 0.99 | 0.99 |
| Discount for cost $\gamma_C$ | 0.99 | 0.99 | 0.99 | 0.99 | 0.99 |
| Batch size | 2048 | 2048 | 2048 | 2048 | 2048 |
| Minibatch size | 64 | 64 | N/A | N/A | 64 |
| No. of optimization epochs | 10 | 10 | N/A | N/A | 10 |
| Maximum episode length | 1000 | 1000 | 1000 | 1000 | 1000 |
| GAE parameter (reward) | 0.95 | 0.95 | 0.95 | 0.95 | 0.95 |
| GAE parameter (cost) | N/A | 0.95 | 0.95 | 0.95 | 0.95 |
| Learning rate for policy | $3 \times 10^{-4}$ | $3 \times 10^{-4}$ | N/A | N/A | $3 \times 10^{-4}$ |
| Learning rate for reward value net | $3 \times 10^{-4}$ | $3 \times 10^{-4}$ | $3 \times 10^{-4}$ | $3 \times 10^{-4}$ | $3 \times 10^{-4}$ |
| Learning rate for cost value net | N/A | $3 \times 10^{-4}$ | $3 \times 10^{-4}$ | $3 \times 10^{-4}$ | $3 \times 10^{-4}$ |
| Learning rate for $\nu$ | N/A | 0.01 | 0.01 | N/A | 0.01 |
| $L2$-regularization coeff. for value net | $3 \times 10^{-3}$ | $3 \times 10^{-3}$ | $3 \times 10^{-3}$ | $3 \times 10^{-3}$ | $3 \times 10^{-3}$ |
| Clipping coefficient | 0.2 | 0.2 | N/A | N/A | N/A |
| Damping coeff. | N/A | N/A | 0.01 | 0.01 | N/A |
| Backtracking coeff. | N/A | N/A | 0.8 | 0.8 | N/A |
| Max backtracking iterations | N/A | N/A | 10 | 10 | N/A |
| Max conjugate gradient iterations | N/A | N/A | 10 | 10 | N/A |
| Iterations for training value net[1] | 1 | 1 | 80 | 80 | 1 |
| Temperature $\lambda$ | N/A | N/A | N/A | N/A | 1.5 |
| Trust region bound $\delta$ | N/A | N/A | 0.01 | 0.01 | 0.02 |
| Initial $\nu$, $\nu_{\max}$ | N/A | 0, 1 | 0, 2 | N/A | 0, 2 |

where $x, y$ are the positions of the agent on the plane, $v_x, v_y$ are the velocities of the agent along the $x$ and $y$ directions, $r$ is the radius of the circle, and $x_{\lim}$ specifies the range of the safety region. The radius is set to $r = 10$ for both Ant and Humanoid while $x_{\lim}$ is set to 3 and 2.5 for Ant and Humanoid respectively. Note that these settings are identical to those of the circle task in Achiam et al. (2017). Our experiments were implemented in OpenAI Gym (Brockman et al., 2016) while the circle tasks in Achiam et al. (2017) were implemented in rllab (Duan et al., 2016). We also excluded the Point agent from the original experiments since it is not a valid agent in OpenAI Gym. The first two dimensions in the state space are the $(x, y)$ coordinates of the center mass of the agent, hence the state space for both agents has two extra dimensions compared to the standard Ant and Humanoid environments from OpenAI Gym. Our open-source implementation of the circle environments can be found at `https://github.com/ymzhang01/mujoco-circle`.

### G.2.2 Algorithmic Hyperparameters

For these tasks, we used identical settings as the robots with speed limit tasks except we used a batch size of 50000 for all algorithms and a minibatch size of 1000 for PPO, PPO-Lagrangian, and FOCOPS. The discount rate for both reward and cost were set to 0.995. For FOCOPS, we set $\lambda = 1.0$ and $\delta = 0.04$.

## H    Generalization Analysis

We used trained agents using all four algorithms (PPO Lagrangian, TRPO Lagrangian, CPO, and FOCOPS) on robots with speed limit tasks shown in Figure 1. For each algorithm, we picked the seed with the highest maximum return of the last 100 episodes which does not violate the cost constraint at the end of training. The reasoning here is that for a fair comparison, we wish to pick the best performing seed for each algorithm. We then ran 10 episodes using the trained agents on 10

unseen random seeds (identical seeds are used for all four algorithms) to test how well the algorithms generalize over unseen data. The final results of running the trained agents on the speed limit and circle tasks are reported in Tables 4. We note that on unseen seeds FOCOPS outperforms the other three algorithms on five out of six tasks.

Table 4: Average return of 10 episodes for trained agents on the robots with speed limit tasks on 10 unseen random seeds. Results shown are the bootstrap mean and normal $95\%$ confidence interval with 1000 bootstrap samples.

| Environment | | PPO-L | TRPO-L | CPO | FOCOPS |
|---|---|---|---|---|---|
| Ant-v3 | Reward | $920.4 \pm 75.9$ | $1721.4 \pm 191.2$ | $1335.57 \pm 43.17$ | $\mathbf{1934.9 \pm 99.5}$ |
| (103.12) | Cost | $68.25 \pm 11.05$ | $99.20 \pm 2.55$ | $80.72 \pm 3.82$ | $105.21 \pm 5.91$ |
| HalfCheetah-v3 | Reward | $1698.0 \pm 22.5$ | $1922.4 \pm 12.9$ | $1805.5 \pm 60.0$ | $\mathbf{2184.3 \pm 32.6}$ |
| (151.99) | Cost | $150.21 \pm 4.47$ | $179.82 \pm 1.73$ | $164.67 \pm 9.43$ | $158.39 \pm 6.56$ |
| Hopper-v3 | Reward | $2084.9 \pm 39.69$ | $2108.8 \pm 24.8$ | $\mathbf{2749.9 \pm 47.0}$ | $2446.2 \pm 9.0$ |
| (82.75) | Cost | $83.43 \pm 0.41$ | $82.17 \pm 1.53$ | $52.34 \pm 1.95$ | $81.26 \pm 0.88$ |
| Humanoid-v3 | Reward | $582.2 \pm 28.9$ | $3819.3 \pm 489.2$ | $1814.8 \pm 221.0$ | $\mathbf{4867.3 \pm 350.8}$ |
| (20.14) | Cost | $18.93 \pm 0.93$ | $18.60 \pm 1.27$ | $20.30 \pm 1.81$ | $21.58 \pm 0.74$ |
| Swimmer-v3 | Reward | $37.90 \pm 1.05$ | $33.48 \pm 0.44$ | $33.45 \pm 2.30$ | $\mathbf{39.37 \pm 2.04}$ |
| (24.52) | Cost | $25.49 \pm 0.57$ | $32.81 \pm 2.61$ | $22.61 \pm 0.33$ | $17.23 \pm 1.64$ |
| Walker2d-v3 | Reward | $1668.7 \pm 337.1$ | $2638.9 \pm 163.3$ | $2141.7 \pm 331.9$ | $\mathbf{3148.6 \pm 60.5}$ |
| (81.89) | Cost | $79.23 \pm 1.24$ | $90.96 \pm 0.97$ | $40.67 \pm 6.86$ | $73.35 \pm 2.67$ |

# I  Sensitivity Analysis

We tested FOCOPS across ten different values of $\lambda$, and five difference values of $\nu_{\max}$ while keeping all other parameters fixed by running FOCOPS for 1 millon samples on each of the robots with speed limit experiment. For ease of comparison, we normalized the values by the return and cost of an unconstrained PPO agent trained for 1 million samples (i.e. if FOCOPS achieves a return of $x$ and an unconstrained PPO agent achieves a result of $y$, the normalized result reported is $x/y$) The results on the robots with speed limit tasks are reported in Tables 5 and 6. We note that the more challenging environments such as Humanoid are more sensitive to parameter choices but overall FOCOPS is largely insensitive to hyperparameter choices (especially the choice of $\nu_{\max}$). We also presented the performance of PPO-L and TRPO-L for different values of $\nu_{\max}$.

Table 5: Performance of FOCOPS for Different $\lambda$

| | Ant-v3 | | HalfCheetah-v3 | | Hopper-v3 | | Humanoid-v3 | | Swimmer-v3 | | Walker2d-v3 | | All Environments | |
|---|---|---|---|---|---|---|---|---|---|---|---|---|---|---|
| $\lambda$ | Reward | Cost | Reward | Cost | Reward | Cost | Reward | Cost | Reward | Cost | Reward | Cost | Reward | Cost |
| 0.1 | 0.66 | 0.55 | 0.38 | 0.46 | 0.77 | 0.50 | 0.63 | 0.52 | 0.34 | 0.51 | 0.43 | 0.48 | 0.53 | 0.50 |
| 0.5 | 0.77 | 0.54 | 0.38 | 0.45 | 0.97 | 0.50 | 0.71 | 0.54 | 0.36 | 0.50 | 0.66 | 0.50 | 0.64 | 0.50 |
| 1.0 | 0.83 | 0.55 | 0.47 | 0.47 | 1.04 | 0.50 | 0.80 | 0.52 | 0.34 | 0.49 | 0.76 | 0.49 | 0.70 | 0.50 |
| 1.3 | 0.83 | 0.55 | 0.42 | 0.47 | 1.00 | 0.50 | 0.85 | 0.53 | 0.36 | 0.51 | 0.87 | 0.49 | 0.72 | 0.51 |
| 1.5 | 0.83 | 0.55 | 0.42 | 0.47 | 1.01 | 0.50 | 0.87 | 0.52 | 0.37 | 0.51 | 0.87 | 0.50 | 0.73 | 0.51 |
| 2.0 | 0.83 | 0.55 | 0.42 | 0.47 | 1.06 | 0.50 | 0.89 | 0.52 | 0.37 | 0.52 | 0.82 | 0.45 | 0.73 | 0.51 |
| 2.5 | 0.79 | 0.54 | 0.43 | 0.47 | 1.03 | 0.50 | 0.94 | 0.53 | 0.35 | 0.50 | 0.73 | 0.49 | 0.71 | 0.51 |
| 3.0 | 0.76 | 0.54 | 0.42 | 0.47 | 1.01 | 0.49 | 0.92 | 0.52 | 0.41 | 0.50 | 0.77 | 0.49 | 0.72 | 0.50 |
| 4.0 | 0.70 | 0.54 | 0.40 | 0.46 | 1.00 | 0.49 | 0.87 | 0.53 | 0.43 | 0.49 | 0.64 | 0.49 | 0.67 | 0.50 |
| 5.0 | 0.64 | 0.55 | 0.40 | 0.47 | 1.01 | 0.50 | 0.81 | 0.54 | 0.38 | 0.49 | 0.57 | 0.50 | 0.63 | 0.51 |

Table 6: Performance of FOCOPS for Different $\nu_{\max}$

| $\nu_{\max}$ | Ant-v3 | | HalfCheetah-v3 | | Hopper-v3 | | Humanoid-v3 | | Swimmer-v3 | | Walker2d-v3 | | All Environments | |
|---|---|---|---|---|---|---|---|---|---|---|---|---|---|---|
| | Reward | Cost | Reward | Cost | Reward | Cost | Reward | Cost | Reward | Cost | Reward | Cost | Reward | Cost |
| 1 | 0.83 | 0.55 | 0.45 | 0.61 | 1.00 | 0.51 | 0.87 | 0.52 | 0.40 | 0.62 | 0.88 | 0.50 | 0.74 | 0.55 |
| 2 | 0.83 | 0.55 | 0.42 | 0.47 | 1.01 | 0.50 | 0.87 | 0.52 | 0.35 | 0.51 | 0.87 | 0.50 | 0.73 | 0.51 |
| 3 | 0.81 | 0.54 | 0.41 | 0.47 | 1.01 | 0.49 | 0.83 | 0.53 | 0.34 | 0.49 | 0.87 | 0.50 | 0.71 | 0.50 |
| 5 | 0.82 | 0.55 | 0.41 | 0.47 | 1.01 | 0.50 | 0.83 | 0.53 | 0.31 | 0.49 | 0.87 | 0.50 | 0.71 | 0.51 |
| 10 | 0.82 | 0.55 | 0.41 | 0.47 | 1.01 | 0.50 | 0.83 | 0.53 | 0.34 | 0.47 | 0.87 | 0.50 | 0.71 | 0.50 |
| $+\infty$ | 0.82 | 0.55 | 0.41 | 0.47 | 1.01 | 0.50 | 0.83 | 0.53 | 0.35 | 0.47 | 0.88 | 0.50 | 0.72 | 0.50 |

Table 7: Performance of PPO Lagrangian for Different $\nu_{\max}$

| $\nu_{\max}$ | Ant-v3 | | HalfCheetah-v3 | | Hopper-v3 | | Humanoid-v3 | | Swimmer-v3 | | Walker2d-v3 | | All Environments | |
|---|---|---|---|---|---|---|---|---|---|---|---|---|---|---|
| | Reward | Cost | Reward | Cost | Reward | Cost | Reward | Cost | Reward | Cost | Reward | Cost | Reward | Cost |
| 1 | 0.80 | 0.55 | 0.41 | 0.49 | 0.98 | 0.49 | 0.73 | 0.52 | 0.28 | 0.50 | 0.77 | 0.50 | 0.66 | 0.51 |
| 2 | 0.71 | 0.49 | 0.36 | 0.50 | 0.81 | 0.48 | 0.73 | 0.52 | 0.32 | 0.50 | 0.72 | 0.50 | 0.61 | 0.50 |
| 3 | 0.78 | 0.54 | 0.36 | 0.47 | 0.73 | 0.49 | 0.73 | 0.52 | 0.40 | 0.48 | 0.72 | 0.50 | 0.62 | 0.50 |
| 5 | 0.77 | 0.53 | 0.35 | 0.47 | 0.73 | 0.49 | 0.73 | 0.52 | 0.40 | 0.49 | 0.72 | 0.50 | 0.62 | 0.50 |
| 10 | 0.77 | 0.54 | 0.36 | 0.47 | 0.73 | 0.49 | 0.73 | 0.52 | 0.40 | 0.49 | 0.72 | 0.50 | 0.62 | 0.50 |
| $+\infty$ | 0.66 | 0.54 | 0.27 | 0.45 | 0.73 | 0.49 | 0.55 | 0.47 | 0.40 | 0.49 | 0.72 | 0.50 | 0.55 | 0.49 |

Table 8: Performance of TRPO Lagrangian for Different $\nu_{\max}$

| $\nu_{\max}$ | Ant-v3 | | HalfCheetah-v3 | | Hopper-v3 | | Humanoid-v3 | | Swimmer-v3 | | Walker2d-v3 | | All Environments | |
|---|---|---|---|---|---|---|---|---|---|---|---|---|---|---|
| | Reward | Cost | Reward | Cost | Reward | Cost | Reward | Cost | Reward | Cost | Reward | Cost | Reward | Cost |
| 1 | 0.71 | 0.50 | 0.70 | 0.68 | 0.61 | 0.50 | 0.68 | 0.50 | 0.43 | 0.61 | 0.48 | 0.50 | 0.61 | 0.55 |
| 2 | 0.70 | 0.51 | 0.50 | 0.53 | 0.39 | 0.53 | 0.68 | 0.50 | 0.33 | 0.53 | 0.36 | 0.50 | 0.49 | 0.52 |
| 3 | 0.70 | 0.51 | 0.52 | 0.53 | 0.41 | 0.53 | 0.68 | 0.50 | 0.30 | 0.67 | 0.35 | 0.50 | 0.49 | 0.54 |
| 5 | 0.70 | 0.51 | 0.49 | 0.52 | 0.36 | 0.52 | 0.68 | 0.50 | 0.23 | 0.67 | 0.35 | 0.51 | 0.47 | 0.54 |
| 10 | 0.70 | 0.51 | 0.48 | 0.51 | 0.34 | 0.52 | 0.68 | 0.50 | 0.31 | 0.77 | 0.34 | 0.50 | 0.47 | 0.55 |
| $+\infty$ | 0.70 | 0.51 | 0.48 | 0.51 | 0.36 | 0.52 | 0.68 | 0.50 | 0.30 | 0.78 | 0.34 | 0.50 | 0.48 | 0.55 |

## Footnotes

[1] for PPO, PPO-L, and FOCOPS, this refers to the number of iteration for training the value net per minibatch update.