[Reviews · NeurIPS 2020]

Review 1

Summary and Contributions: The authors present FOCOPS, a first-order method for solving constrained MDPs, with theoretical guarantees about the magnitude of violations during training. The method adopts a two-step approach. First a constrained optimisation is performed in the space of non-parametric policies that maximises expected advantages subject to constraints on the change w.r.t. the current parametric policy as well as the expected cost. Secondly, this non-parametric policy is projected back to the space of parametric policies. In order to use gradient descent in practice, the gradient of both steps w.r.t. the parameters are combined in a single analytical expression. A number of simplifications are made to make the approach easier to implement. Experiments on a number of speed-limited locomotion tasks show the efficacy the the proposed approach, achieving higher task reward while satisfying the cost constraint compared to three baselines. The authors also evaluate the sensitivity and generalisation capabilities of their approach.

Strengths: 1. A novel approach for solving constrained MDPs, an important subfield of reinforcement learning for safety, robotics, etc., that moreover is easy to implement. This work is very relevant as it's becoming clear that scalar rewards are often not sufficient for real-world applications. 2. Well-founded claims, incl. formal proof of the proposed approach. 3. Very thorough evaluation, with well-chosen baselines, proper statistical measures and analysis of generalisation and sensitivity.

Weaknesses: 1. There are a number of simplifications and tweaks being made for the practical implementation (e.g. the indicator function), which may potentially impact any theoretical guarantees. A brief discussion on this would be appreciated, as well as an empirical evaluation. 2. The chosen environments for the experiments are perhaps not the easiest to interpret. For one, in the speed-limited locomotion tasks, it's unclear what is actually being optimised as the speed is also a part of the reward, which makes reward and cost significantly antagonistic. What else is being optimised for?

Correctness: The authors provide formal proofs of their derivations, as well as proper statistical measures for their evaluation, so I would deem both the method and experiments correct.

Clarity: The paper is very well written, with a concise notation, and sufficient motivation for most of the choices made. Proofs are sufficiently detailed. Minor comment: adding the cost constraints to the tables instead of just the supplementary material would make it more clear how much they are (not) violated.

Relation to Prior Work: While brief, the authors do describe and compare to the main recent CMDP algorithms.

Reproducibility: Yes

Additional Feedback: A few additional comments: 1. The implementation of the PPO/TRPO-L baselines is not entirely clear, specifically what the gradient of the cost w.r.t. the parameters looks like. Ignoring the KL constraint in FOCOPS & TRPO, the gradient w.r.t. reward and cost terms may potentially look very similar between both algorithms. 2. While the sensitivity analysis is much appreciated, it would also have been good to include the delta parameter from the practical implementation, as this seems to somewhat serve the purpose of an adaptive lambda (used in the indicator function and termination condition for the inner loop), and may suppress the impact of different fixed values for lambda. 3. For the sensitivity analysis, it is unclear what a (unique) random seed influences. Does it correspond to a fixed initial configuration? UPDATE I would like to thank the authors for their additional clarifications, and stay with my decision to accept this paper.


Review 2

Summary and Contributions: This paper proposes an algorithm which maximizes an RL agent's reward while satisfying certain constraints. The method solves a constrained optimization problem in a non-parameterized policy space and then projects the update policy into the space of parameterized policies.

Strengths: - FOCOPS eliminates error in CPO from Taylor approximations and approximating the inverse of a Fisher information matrix. - The method is simple to implement relative to prior approaches, since it only uses first-order approximations. - FOCOPS updates the policy to improve performance while satisfying constraints during training. - Experiments demonstrate that FOCOPS surpasses performance of prior constrained policy optimization methods while satisfying constraints. - FOCOPS is not particularly sensitive to the choice of its hyper-parameters lambda and v_max.

Weaknesses: - Only one constraint value is selected for each environment (50% of the speed attained by an unconstrained PPO agent - how and why was 50% chosen?). I would be interested in seeing whether FOCOPS consistently exceeds performance and satisfies constraints compared to other methods for different constraint levels. - Only one constraint is used for the experiments. How would FOCOPS perform empirically when there are multiple constraints?

Correctness: Yes.

Clarity: The paper is clear and easy to follow.

Relation to Prior Work: Yes, this paper's relation to prior work is discussed in detail.

Reproducibility: Yes

Additional Feedback: I didn't fully understand why the per-state acceptance indicator function is added. If D_KL(\pi_\theta || \pi_{\theta_k})[s] is large wouldn't that in turn increase D_KL(\pi_\theta || \pi^*), meaning that you would want to sample those states more in order to successfully minimize D_KL(\pi_\theta || \pi^*)? For the experiments, I think it would be helpful to list the cost thresholds for each environment in the tables to show clearly whether or not the cost constraint is being satisfied by each method. Update (post rebuttal): I thank the authors for addressing my comments in the rebuttal. I have kept my score of 7 and recommend acceptance.


Review 3

Summary and Contributions: This paper proposed a new approach for constrained RL problems: First Order Constrained Optimization in Policy Space (FOCOPS), which optimizes the expected discounted reward with the guarantee to satisfy a set of constraints represented by bounded expected costs. This paper follows the same problem formulation as CPO but the overall solution is much simpler. Instead of approximating the original optimization problem as a quadratic programming problem that requires a solver, FOCOPS is basically doing gradient descent with the help of several key approximations, which is simpler both conceptually and (very likely) in practice. The authors showed with numeric experiments that FOCOPS can both approximately satisfy the constraints during training and outperform the other baselines in objective in multiple cases. I would like to keep all my comments the same after reading the authors' rebuttal, as I found nothing unexpected there.

Strengths: The idea behind this work is very clear. Theorem 1 can be easily derived using duality, but the difficulty lies in how to solve it. The paper is creative in its simplification on the update of dual varaibles lambda and mu. Without this, the algorithm would be pretty much a primal-dual gradient method, which requires the primal part to be evaluated accurately enough before the update of dual variables, which is difficult in practice. For lambda, the paper takes advantage of its similarity to the temperature term in maximum entropy RL and thus fixes it during training. For mu, it takes the advantage that the current policy is close to the optimal one. These simplifications are both intuitive and effective. The empirical experiments are on MuJoCo robots and the circle experiment from the CPO paper. All the baselines (PPO-L, TRPO-L, CPO) follow similar ideas for constrained RL problems and thus are directly comparable. The performances of FOCOPS are quite consistent over different experiments. The authors also did some sensitivity analysis to show that their approximations on lambda and mu_max make sense in these experiments. The topic of constrained RL is of great concern to the NeurIPS community.

Weaknesses: The success of this method would be dependent on the effectiveness of several approximations, which poses some assumptions on the problem. For example, the loss is not sensitive to lambda. It is possible that these approximations are not effective on some other applications. But we should also admit that there are no perfect solutions for such a complicated general problem. Also, the algorithm replies on getting new samples from the current policy in each iteration and thus is expected to have high sample complexity. It would be interesting to see if it is possible to combine off-policy data into the optimization procedure. Again, this is difficult and can be addressed in separate future works.

Correctness: I did not check the detailed proofs in supplemental material, but the ideas behind the theoretical part are straightforward. The empirical methodology is similar to that of the CPO paper and also makes sense to me.

Clarity: This paper is well written and easy to follow, especially if the readers are familiar with the CPO paper. My only complaint is on Table 1 and 2: It is hard to tell how well the learned policies satisfy the constraints, as the feasibility bounds are not listed.

Relation to Prior Work: This paper has covered the most relevant papers that I know such as CPO, Chow's primal-dual method and PCPO. The differences between this paper and these previous works are well explained. Although there are definitely more works on safe reinforcement learning (such as safe exploration, lyapunov-based methods, etc), I don't think there will be enough space to have an overview of all the major ideas.

Reproducibility: No

Additional Feedback: I like this paper for its simplicity. I just have a few comments here: * For Table 1 and 2: It would be helpful to list the boundaries of cost functions in the table. * Some comments would be welcome to address the case when the initial policy is infeasible. Will that make any difference to the derivation of the algorithm? I saw that the Swimmer-v3 task started from infeasible policies but it seems that most methods recovered from it eventually. * It would be great if the authors can provide an open-source toolbox for FOCOPS such that people can reproduce the fantastic results here. There can be important implementation details that are otherwise neglected. * It would be interesting if the authors can discuss about the limitations of this work.


Review 4

Summary and Contributions: This work proposes a new RL method for solving Constrained Markov Decision Processes (CMDPs). This approach builds on the theory proposed by Achiam et al. (2017) but uses an alternative (simpler) method to solve Achiam et al.'s optimization problem (1-3). It first finds a non-parametric policy that optimally solves (1-3) and then projects that policy into the parameterized policy space by finding the closest parameterized policy to the non-parametric solution. A distinctive advantage of this approach is its simplicity (both subproblems can be solved using first-order approximations). Experiments in MuJoCo environments show promising results w.r.t. three strong baselines. The paper's main contributions are 1) a near-closed form solution for (1-3) in the space of nonparametric policies and 2) a set of first-order approximations to derive a practical approach to solve CMDPs, called FOCOPS.

Strengths: 1. The idea of solving the non-parametric version of the optimization problem (1-3) is interesting and, as far as I know, novel. 2. The derivation of FOCOPS is mathematically sound. 3. FOCOPS has strong empirical performance.

Weaknesses: 1. If I understood correctly, the optimal nonparametric solution is guaranteed to satisfy the cost constraints. However, once the nonparametric solution is projected to the parametric space, there are no guarantees w.r.t. satisfying those constraints. Is this correct? If so, wouldn't FOCOPS be potentially more unsafe than alternative approaches, such as CPO or PCPO, for some (possibly adversarial) environments? 2. Similarly, it is unclear to me the relationship between the resulting policy \pi_theta from solving (4-6) and then (10) vs the actual optimal solution of (1-3). I think the paper would benefit from adding a discussion about this.

Correctness: The theory and experiments in the paper look correct.

Clarity: The paper is self-contained and well written. I only spotted a small typo in line 202.

Relation to Prior Work: Previous works are well-covered. Unfortunately, this work does not include an empirical comparison with Yang et al. (2020)---which also decomposes the optimization problem into two subproblems. Are there theoretical (or practical) reasons to prefer FOCOPS over PCPO (beyond being potentially simpler to implement)?

Reproducibility: Yes

Additional Feedback: - An interesting advantage of FOCOPS is its simplicity w.r.t. previous approaches. Is it also faster than existing methods? - I encourage the authors to release their code. - Consider citing this recently published work: Stooke, Adam, Joshua Achiam, and Pieter Abbeel. "Responsive Safety in Reinforcement Learning by PID Lagrangian Methods." arXiv preprint arXiv:2007.03964 (2020). ----------------------- Post-rebuttal ----------------------- My concerns have been addressed and I recommend acceptance.

[Author Response · NeurIPS 2020]

We would like to begin by thanking all the reviewers for their hard work in providing us with such insightful feedback. We are encouraged that the reviewers found our work to be simple and intuitive (**R1**, **R2**, **R3**), novel (**R1**, **R5**), and easy to implement (**R1**,**R2**). Several reviewers have also given credit to our work for sound theoretical claims/proofs (**R1**, **R5**) and thorough empirical evaluation (**R1**, **R3**). We will also be releasing the code for our implementation as it is our hope that our method not only provides a simple baseline for comparing new algorithms in constrained RL, but moreover makes it more accessible for researchers from other fields to apply RL to their own work.

Several reviewers noted that the guarantee in (9) may no longer hold post-approximation (**R1**, **R3**, **R5**). **R3** also pointed out that these approximations may prove to be ineffective in other applications. We acknowledge that these are valid concerns, but would also like to point out that the same can be said for most DRL methods. Our superior empirical performance compared to previous works show that the approximations we make are less destructive. As such, we do not believe our algorithm is less safe compared to CPO/PCPO (**R5**) which also makes extensive approximations. Furthermore, we are grateful to **R3** for noting that our approximations are both intuitive and effective.

We would like also to clarify the use of the indicator function in response to **R1** and **R2**. The indicator function enforces the constraint that $\pi_\theta$ is not too far from $\pi_{\theta_k}$. This is also important because our method is a first-order method, so the approximations that we make is only accurate near the initial condition (i.e. $\pi_\theta = \pi_{\theta_k}$). We enforce this condition by ensuring $D_{KL}(\pi_\theta \parallel \pi_{\theta_k})$ do not diverge too much. The large distance between $\pi_\theta$ and $\pi_{\theta_k}$ doesn't mean that the distance between $\pi_\theta$ and $\pi^*$ is large. At iteration $k$, before we make any update, $\pi_\theta = \pi_{\theta_k}$. As we make more gradient updates during iteration $k$, we expect $\pi_\theta$ to diverge from $\pi_{\theta_k}$ while becoming closer $\pi^*$. That is, the distance between $\pi_\theta$ and $\pi_{\theta_k}$ is *increasing*, but the distance between $\pi_\theta$ and $\pi^*$ is *decreasing*.

Several reviewers recommended adding the constraint threshold values to the tables (**R1**, **R2**, **R3**). This will be done in our revision of the paper. Finally we would like to address other comments/concerns made by the reviewers.

**R1** 1) The goal of the MuJoCo environments is to train the agents to walk as fast as possible without falling over while not overexerting the joints. Hence the reward consists of multiple term which takes into account all such aspects. Our environment imposes a speed limit on the agents (which is reasonable in a safety-constrained setting) thus our policy forces the agent to optimize for the other terms in the reward (such as "stability" and torque applied to joints) while controlling for the speed. 2) In our subsequent revision we will explicitly write out the gradient terms for both PPO-L and TRPO-L. The reviewer is right in that the gradient for the cost term is very similar but the reward gradient term differ significantly since TRPO is a second-order method. 3) In our experiments, the random seeds determine both the initial weights of the neural nets and initial configuration of the environments.

**R2** 1) We have not had the opportunity to experiment on different constraint thresholds but we agree with the reviewer that these results would be interesting to see. We will be running these experiments and including the results in our subsequent revision. 2) In theory it is possible to extend FOCOPS to multiple constraints by introducing additional dual variables, we focused on the one constraint case since it results in cleaner maths and easier to interpret experiments. In our revision, we will make clearer the scope of our paper (single constraint). While FOCOPS like most similar work such as CPO focused on a single constraint, we concur that the multi-constraint case deserves further research.

**R3** 1) FOCOPS is an on-policy algorithm hence it inherits many of its flaws such as high sample complexity. We thank the reviewer for pointing this out. In our experience, learning constraint-satisfying policies from off-policy data is extremely challenging and deserves further research. 2) The reviewer is correct in pointing out that the theory of both FOCOPS and CPO assume the initial policy to be feasible. However in practice, the gradient update term increases the dual variable associated with the cost when the cost constraint is violated, this would result in a feasible policy after a certain number of iterations. We also observed that this is indeed the case with the swimmer environment.

**R5** 1) It is in general not computationally feasible to solve (1-3) directly therefore it would be difficult to compare its solution to FOCOPS. However it is possible to compare the gradient update term for CPO, which uses a second order approximation of (1-3) and the gradient update term for FOCOPS. We will add a brief discussion on this in our subsequent revision. 2) While we appreciate the novelty of PCPO's alternative two-step solution, empirically speaking PCPO does not seem to consistently beat CPO based on results reported in the original paper. To quote one of the meta-reviewers for PCPO from ICLR 2020: "The experimental evidence is a bit mixed, with the best of the proposed projections (based on the KL approach) sometimes beating CPO but also sometimes being beaten by it, both on the obtained reward and on constraint satisfaction". In contrast, FOCOPS outperformed CPO on all test environments. 3) In terms of computational speed, CPO takes one large gradient step while FOCOPS combines many smaller gradient steps using minibatches with early stopping. Due to the larger number of gradient steps, FOCOPS is in general slightly slower than CPO on most environments. However we found this difference to be marginal. 4) We would like to thank the reviewer for pointing us to the recent ICML 2020 paper from Stooke et al. and will add a brief discussion in our subsequent revision.

[Meta-Review · NeurIPS 2020]

The reviewers were unequivocally positive about this paper. They had some minor concerns that the authors address in their response. I encourage the authors to incorporate this into any final version.